# IN SEARCH OF FORGOTTEN DOMAIN GENERALIZATION

**Prasanna Mayilvahanan**[1,2,3*]   **Roland S. Zimmermann**[1,2,3*]   **Thaddäus Wiedemer**[1,2,3]

**Evgenia Rusak**[1,2,3]   **Attila Juhos**[1,2,3]   **Matthias Bethge**[1,2]   **Wieland Brendel**[2,3,4]

## ABSTRACT

Out-of-Domain (OOD) generalization is the ability of a model trained on one or more domains to generalize to unseen domains. In the ImageNet era of computer vision, evaluation sets for measuring a model's OOD performance were designed to be strictly OOD with respect to style. However, the emergence of foundation models and expansive web-scale datasets has obfuscated this evaluation process, as datasets cover a broad range of domains and risk test domain contamination. In search of the forgotten domain generalization, we create large-scale datasets subsampled from LAION—LAION-Natural and LAION-Rendition—that are strictly OOD to corresponding ImageNet and DomainNet test sets in terms of style. Training CLIP models on these datasets reveals that a significant portion of their performance is explained by in-domain examples. This indicates that the OOD generalization challenges from the ImageNet era still prevail and that training on web-scale data merely creates the illusion of OOD generalization. Furthermore, through a systematic exploration of combining natural and rendition datasets in varying proportions, we identify optimal mixing ratios for model generalization across these domains. Our datasets and results re-enable meaningful assessment of OOD robustness at scale—a crucial prerequisite for improving model robustness.

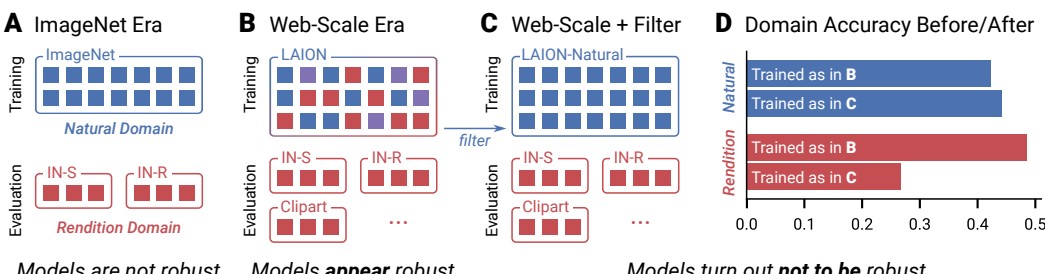

Figure 1: **Evaluated correctly, CLIP's OOD performance on renditions drops significantly**. **A**: Models used to be trained on a single domain like *natural images* from ImageNet (Russakovsky et al., 2015) and evaluated for out-of-domain (OOD) generalization on a different domain like *renditions* from test sets such as ImageNet-R (Hendrycks et al., 2021a), ImageNet-Sketch (Wang et al., 2019). **B**: Today, large foundation models like CLIP (Radford et al., 2021) are trained on web-scale datasets such as LAION-400M (Schuhmann et al., 2021) containing images from many domains. Tested on a specific domain like renditions, CLIP exhibits unprecedented performance and appears robust. **C**: We subsample from a deduplicated LAION-400M (Abbas et al., 2023) to obtain LAION-Natural, a web-scale dataset containing only natural images, which re-enables a meaningful assessment of CLIP's generalization performance to renditions. **D**: CLIP trained on LAION-Natural performs noticeably poorer on renditions, suggesting that its OOD performance has been previously overestimated. The models are evaluated on refined test datasets containing samples only from their intended domains.

---

*Equal contribution. [1]University of Tübingen, [2]Tübingen AI Center, [3]Max-Planck-Institute for Intelligent Systems, Tübingen, [4]ELLIS Institute Tübingen. Contact: prasanna.mayilvahanan@uni-tuebingen.de, research@rzimmermann.com. Code available at https://brendel-group.github.io/clip-dg/.

# 1    INTRODUCTION

Foundation models have revolutionized our world, demonstrating remarkable capabilities in solving grade school math problems, writing creative essays, generating stunning images, and comprehending visual content (OpenAI, 2023; Schulman et al., 2022; Ramesh et al., 2022). One notable example is CLIP (Radford et al., 2021), a vision-language model pretrained on a vast dataset of image-text pairs, which forms the backbone of numerous other foundation models (Ramesh et al., 2022; Liu et al., 2023a). CLIP has achieved unprecedented performance across a wide range of benchmarks spanning many domains—a sharp contrast to models from the ImageNet era, which struggled to generalize from a training domain mostly consisting of natural photographs to stylistically different domains such as ImageNet-Sketch (Wang et al., 2019), ImageNet-R (Hendrycks et al., 2021a), and DomainNet (Peng et al., 2019).

Domains, while often challenging to quantify in practice (Ben-David et al., 2010), emerge from collecting data from specific sources and conditions. Some domains, like *natural images* or *renditions*, are better delineated, allowing the creation of datasets like the ones mentioned above. Out-of-domain (OOD) generalization refers to a model's ability to perform well on data from domains other than its training domain(s) (Wang et al., 2021). In this work, we collectively refer to the domain represented by ImageNet-Sketch, ImageNet-R, DomainNet-Painting, DomainNet-Clipart, DomainNet-Sketch, and DomainNet-Quickdraw as the *rendition domain*, since it contains images that are renditions of natural objects and scenes. Generalization to the rendition domain (especially OOD) is crucial for aligning models with human perception, as humans can interpret abstract visual renditions, while machines tend to rely heavily on textural cues (Hendrycks et al., 2021a; Geirhos et al., 2019).

CLIP's strong performance in several domains, including renditions, is attributed to its vast training distribution, rather than its contrastive learning objective, language supervision, or dataset size (Fang et al., 2022). However, Fang et al. (2022) do not specify what characteristics of the training distribution drive this performance. CLIP could be learning more robust representations due to the diversity of natural images in its training set—or it may simply have been exposed to many datapoints from the (assumed to be OOD) test domains during training. Indeed, Mayilvahanan et al. (2023) revealed that CLIP's training data contains exact or near duplicates of samples of many OOD datasets. Yet, they showed that CLIP still generalizes well when this *sample contamination* is corrected. However, their analysis failed to account for *domain contamination*.

In contrast to sample contamination, domain contamination does not focus on duplicates of specific datapoints but rather examines whether critical aspects of a test domain are present in the training domain, such as images with different content but similar style to test samples. For example, after the correction by Mayilvahanan et al. (2023), many other *rendition* images, while not duplicates, remained in CLIP's training set (refer to Tab. 2). Prior works often assume that CLIP is capable of generalizing OOD (Radford et al., 2021; Abbasi et al., 2024; Nguyen et al., 2024; Fang et al., 2022; Li et al., 2023; Shu et al., 2023); however, it remains unclear whether this is truly the case or if its performance is primarily driven by training on images from the test domain. This leads us to our central question:

*To what extent does domain contamination explain CLIP's performance on renditions?*

We address the central question with the following contributions:

- **Constructing Clean Single-Domain Datasets**: To rigorously test whether CLIP's success in the rendition domain stems from their exposure during training, we first train a domain classifier to distinguish natural images from renditions (Sec. 3.2). By applying the domain classifier to a deduplicated version of LAION-400M, we create and release two datasets: LAION-Natural contains $57\,\text{M}$ natural images; LAION-Rendition consists of $16\,\text{M}$ renditions of scenes and objects. Additionally, we refine existing rendition OOD benchmarks (ImageNet-R, ImageNet-Sketch, etc.) by removing samples that do not belong to the corresponding domain (Sec. 3.4).

- **Refining the Evaluation of CLIP's OOD Performance**: Using LAION-Natural, we demonstrate that CLIP trained only on natural images significantly underperforms on rendition domain shifts (Sec. 4). This suggests that its original success stems from domain contamination, not from an intrinsic OOD generalization ability (see Fig. 1 for a summary).

- **Investigating Domain Mixing and Scaling Effects**: Our single-domain datasets enable analyzing the effects of training on controlled mixtures of natural and rendition images across scales (Sec. 5). We identify the optimal mixing ratio for the best overall performance and show the degree to which training on one domain enables some generalization to the other.

Through this work, we aim to shed light on the limitations of foundation models like CLIP in handling OOD generalization and provide valuable datasets and tools to the community for further exploration. Fig. 1 illustrates our core methodology.

## 2 RELATED WORK

**Measuring the OOD Generalization of CLIP Models** We aim to understand the OOD generalization capabilities of CLIP from a data-centric viewpoint. While multi-modal training with rich language captions does seem to contribute to robustness against distribution shifts (Xue et al., 2024), Fang et al. (2022) demonstrated that the nature of CLIP's training distribution (as opposed to its mere size, its specific training objective, or natural language supervision) causes strong performance on various distribution shifts. However, it is unclear what aspects of the data distribution drive the robustness gains. Mayilvahanan et al. (2023) remove images highly similar to the test sets to show that data contamination and high perceptual similarity between training and test data do not explain generalization performance. While their data pruning technique removes some samples from LAION-400M that lie outside the natural image domain, they do not address domain generalization: They only account for the part of a domain covered by existing test sets and give no guarantee that all images of a given domain were removed. In another line of work, Nguyen et al. (2022) discover that a model's effective robustness (Fang et al., 2022; Taori et al., 2020) on a test set interpolates when training data is compiled from various sources. However, they only consider mixing datasets that each cover multiple domains. In this work, we take their analysis further and show how mixing two data sources from distinct domains interpolates the effective robustness on those domains. Our study's title is inspired by Gulrajani & Lopez-Paz (2021), who studied generalization from multiple distinct source domains. In contrast, we focus on generalization from single or mixed source domains to unseen domains. Overall, we aim for our work to be a valuable addition to the literature on OOD generalization (Liu et al., 2023b; Koh et al., 2021; Madan et al., 2021; Gulrajani & Lopez-Paz, 2021; Madan et al., 2022; Arjovsky et al., 2019; Arjovsky, 2021).

**Domain Classification** The primary goal of our work necessitates creating web-scale datasets of different domains. This entails building a robust domain classifier that can reliably distinguish *natural images* from *renditions*. This task can be regarded as classifying the style of an image, which Gatys et al. (2015) proposed to measure using Gram matrices and which has been widely explored since then (Sandoval et al., 2019; Menis-Mastromichalakis et al., 2020; Sandoval Rodriguez et al., 2018; Joshi et al., 2020; Garcia & Vogiatzis, 2018; Chu & Wu, 2018; Bai et al., 2021). More recently, Cohen-Wang et al. (2024a) use a fine-tuned CLIP model from OpenCLIP (Ilharco et al., 2021) to distinguish between ImageNet and test sets with a domain shift, such as ImageNet-Sketch, ImageNet-R, and ImageNet-V2 (Recht et al., 2019). Wang et al. (2023) and Somepalli et al. (2024) develop a dataset classifier using a backbone trained by self-supervised learning and classification through retrieval via a database. Liu & He (2024) report high performance when training image classifiers to distinguish between different large-scale and diverse datasets.

## 3 CONSTRUCTING CLEAN SINGLE-DOMAIN DATASETS

To answer our central question—how much of CLIP's performance on renditions can be explained by domain contamination—we must filter out datapoints from specific domains within web-scale datasets. Similar to how ImageNet is compared to ImageNet-Sketch and ImageNet-R, or how DomainNet-Real is compared to DomainNet-Sketch (Quickdraw, Infograph, Clipart, and Painting), we aim to create clean *natural* and *rendition* datasets from LAION by building a domain classifier to distinguish between these domains. To build a robust domain classifier, we first create a labeled dataset where each class represents a distinct domain. The labeling process is outlined in Sec. 3.1, and we explore different ways to build a domain classifier in Sec. 3.2. Further, in Sec. 3.3, we employ the best-performing classifiers to analyze the composition of different training and test sets and finally use it to subsample LAION-Natural and LAION-Rendition in Sec. 3.4.

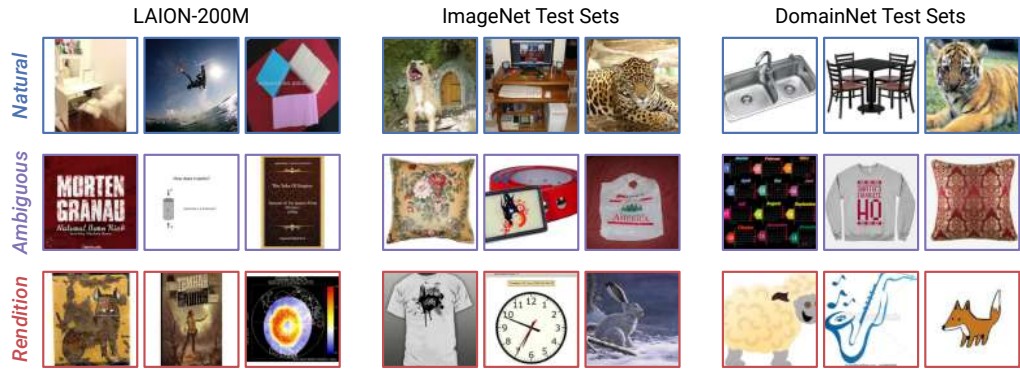

Figure 2: **Labeled *natural*, *ambiguous*, and *rendition* samples from different datasets**. *Natural* images are photos or high-quality renders with minor filters that preserve *fine-grained textures*, while *renditions* are typically sketches, paintings, or graphics with *flat or simplified textures*. Images with elements of both, such as collages or natural images with large stylized elements, and images mainly containing text are labeled as *ambiguous*.

**LAION-200M**   For the remainder of this work, we substitute LAION-400M with LAION-200M, which we obtain by de-duplicating LAION-400M based on perceptual similarity as introduced by Abbas et al. (2023). Both Abbas et al. (2023) and Mayilvahanan et al. (2023) demonstrate that CLIP trained on LAION-200M obtains comparable downstream performance while greatly reducing the computational burden of training models from scratch and analyzing the dataset.

### 3.1 LABELING

LAION-200M contains diverse images from a multitude of sources. The images vary from naturally occurring to synthetically generated. We encourage the reader to glance at Fig. 20 to get a sense of the dataset and the difficulty of determining the domain of each image. We aim to classify these images mainly as *natural* or *renditions*. We also add an extra *ambiguous* class for images with elements of both domains, images with elements of neither, and edge cases.

We manually label images based on a labeling handbook derived from analyzing the existing OOD test sets, which we outline in Appx. A.1.1. In general, we adopt a *texture-centric* approach to distinguish renditions of a scene or object from their natural depictions. That is, depictions where *fine-grained texture information* is preserved are generally considered *natural*, while depictions with *simplified or flat textures* are considered *renditions*. Fig. 2 illustrates this demarcation on samples from LAION-200M, ImageNet test sets, and DomainNet test sets.

To further ease the labeling procedure, we first build a rough binary classifier by fine-tuning CLIP ViT-L/14 with a linear readout to differentiate between some of the *natural* ImageNet and DomainNet test sets (namely, ImageNet-Val, ObjectNet (Barbu et al., 2019), ImageNet-V2, ImageNet-A (Hendrycks et al., 2021b), and DomainNet-Real) and *rendition* test sets (namely, ImageNet-Sketch, ImageNet-R, DomainNet-Painting, DomainNet-Sketch, and DomainNet-Clipart). We use this classifier to roughly pre-label samples before they are annotated by a human. The annotator verifies and potentially updates the labels for 25 images at a time (see Fig. 7).

Overall, we label 19 000 random images from LAION-200M and 1000 images from each of the ImageNet and DomainNet test sets (12 000 in total). Notably, almost all ImageNet and DomainNet test sets usually assumed to contain only images of a single domain exhibit some domain contamination. We discuss this in detail in Sec. 3.3. Tab. 4 contains a detailed breakdown of labels for each dataset. We show more samples grouped by domain for each dataset in Figs. 23 and 34.

### 3.2 TRAINING AND CHOOSING THE DOMAIN CLASSIFIER

With the domain-labeled dataset, we can train a domain classifier to partition all of LAION-200M into *natural* images, *renditions*, or *ambiguous* images. Since we aim to obtain datasets containing

Table 1: **We chose the best *natural* classifier and the best *rendition* classifier** between a binary classifier based on (DR) (Cohen-Wang et al., 2024b) and a ternary classifier using a linear readout based on fine-tuned CLIP model (FT). All models use CLIP ViT-L/14 pretrained on LAION-2B. We report precision and recall for the *natural* class (top) and *rendition* class (bottom) on ImageNet (IN) and DomainNet (DN) test sets and average performance across all test sets. For each class, we select the classifier with the highest validation-recall.

| cls=*natural* | Val | | Test | | IN-Val | | IN-v2 | | IN-A | | ON | | DN-R | | *Average* | |
|---|---|---|---|---|---|---|---|---|---|---|---|---|---|---|---|---|
| **Model** | P | R | P | R | P | R | P | R | P | R | P | R | P | R | P | R |
| DR-R | 0.98 | 0.08 | 0.72 | 0.08 | 1.00 | 0.00 | 1.00 | 0.00 | 1.00 | 0.00 | 0.95 | 0.20 | 1.00 | 0.00 | 0.95 | 0.05 |
| FT | 0.98 | 0.41 | 0.95 | 0.44 | 1.00 | 0.36 | 0.99 | 0.40 | 1.00 | 0.46 | 0.99 | 0.53 | 1.00 | 0.42 | 0.99 | 0.43 |

| cls=*rendition* | Val | | Test | | IN-R | | IN-S | | DN-S | | DN-Q | | DN-P | | DN-C | | DN-I | | *Average* | |
|---|---|---|---|---|---|---|---|---|---|---|---|---|---|---|---|---|---|---|---|---|
| **Model** | P | R | P | R | P | R | P | R | P | R | P | R | P | R | P | R | P | R | P | R |
| DR-R | 0.98 | 0.35 | 0.98 | 0.41 | 1.00 | 0.60 | 1.00 | 0.71 | 1.00 | 0.74 | 1.00 | 0.33 | 0.99 | 0.60 | 1.00 | 0.65 | 0.98 | 0.39 | 0.99 | 0.53 |
| FT | 0.98 | 0.27 | 0.95 | 0.26 | 1.00 | 0.38 | 1.00 | 0.57 | 1.00 | 0.61 | 1.00 | 0.68 | 1.00 | 0.21 | 1.00 | 0.50 | 1.00 | 0.30 | 0.99 | 0.42 |

only images from a single domain, we need a domain classifier that is as precise as possible. To this end, we train classifiers on 13 000 labeled LAION-200M images, retaining 3000 samples each for a validation and test set. From the domain classification literature discussed in Sec. 2, we evaluate four methods with publicly available code. All methods build on CLIP ViT-L/14 pretrained on LAION-2B, which we choose for its balance between accuracy and inference speed. For brevity, we present the two methods we finally employ here, and refer the reader to Appx. A.1.2 for a detailed description, results, and comparisons with other approaches.

**Density Ratios** Cohen-Wang et al. (2024b) aim to estimate the probability that a given sample is drawn from a reference distribution $p_{\text{ref}}$. Since high dimensional density estimation is challenging, they build a classifier to distinguish between a reference and a shifted distribution and compute the density ratio $\frac{p_{\text{ref}}}{p_{\text{shifted}}}$ which they threshold at $0.2$ to classify a given sample. We deploy their method unchanged to our task. We obtain two binary classifiers, DR-N and DR-R, that distinguish natural from non-natural samples and renditions from non-renditions, respectively.

**Fine-Tuning** We fine-tune pretrained CLIP ViT-L/14's image encoder with a randomly initialized linear readout on the training dataset to obtain a ternary classifier, dubbed FT.

We use the validation set to determine the two best domain classifiers, one for natural images and one for renditions. Since the domain classifier should maximize precision above all else, we set the confidence threshold for each model such that it achieves $98\,\%$ per-class precision. We then pick the classifier with the highest per-class recall to minimize the number of datapoints that are discarded when subsampling LAION-200M to build LAION-Natural and LAION-Rendition. We choose FT, the fine-tuned ternary classifier, and DR-R, the binary classifier using density ratios, to detect natural and rendition images, respectively. We use these classifiers for all subsequent experiments. Tab. 1 reports these models' precision and recall on the *natural* and *rendition* class across ImageNet and DomainNet test sets. For comparison to other methods see Appx. A.1.2. For raw accuracy numbers of all models, which in general are high for most, refer to Tabs. 7 and 8 in Appx. A.1.5. We also assess the quality of the labels and the domain classifier's predictions in Appx. A.1.4, finding them to be robust even in the presence of label noise during training.

## 3.3 Analyzing the Domain Make-Up of Different Datasets

Both ImageNet and DomainNet are web-scraped datasets that were refined through extensive human annotation. In contrast, LAION-400M is obtained purely through web scraping without subsequent human domain filtering. Since human annotators can make mistakes, and LAION-200M's domain composition is inherently unknown, we use our domain classifiers to understand it.

To this end, we deploy the chosen classifiers from Sec. 3.2 and label a sample *ambiguous* if the *natural* and *rendition* classifier disagree. We apply the classifiers both with their strict thresholds at $98\,\%$ validation-precision, which yields a strong lower bound for the number of samples in each domain, as well as with their default thresholds, which yields a more rounded estimate.

Table 2: **Domain composition of training sets.** We apply our *natural* and *rendition* domain classifiers with their strict thresholds at $98\%$ validation-precision to get a lower bound of samples from each domain and with their default thresholds to obtain a more balanced estimate. ImageNet-Train has a much smaller fraction of *rendition* samples than LAION-200M. We also note that 'combined-pruned', the training set from Mayilvahanan et al. (2023) that corrected for test set contamination, still contains a large fraction of renditions.

| Dataset | # Samples | Classifier Precision | | Natural | Ambiguous | Rendition |
| | | Natural | Rendition | | | |
| --- | --- | --- | --- | --- | --- | --- |
| LAION-200M | 199 663 250 | 0.79 | 0.77 | 60.74 % | 25.41 % | 13.86 % |
| | | 0.98 | 0.98 | 28.40 % | 63.70 % | 7.90 % |
| ImageNet-Train | 1 281 167 | 0.79 | 0.77 | 89.20 % | 9.62 % | 1.18 % |
| | | 0.98 | 0.98 | 36.00 % | 63.60 % | 0.40 % |
| combined-pruned | 187 471 515 | 0.79 | 0.77 | 62.98 % | 25.18 % | 11.83 % |
| | | 0.98 | 0.98 | 29.58 % | 64.02 % | 6.40 % |

From Tab. 2, it is clear that LAION-200M contains a considerable portion of strictly rendition images (at least $7.90\%$, corresponding to 16 million images), and potentially many more images with some rendition elements in the ambiguous group. In contrast, for ImageNet, we find a much smaller fraction of renditions (at least $0.4\%$ of samples).

Additionally, we observe that **many evaluation datasets are considerably domain-contaminated** (at least $5\%$ of samples stem from the opposite domain), especially ImageNet-R, DomainNet-Real, DomainNet-Clipart, DomainNet-Painting, and DomainNet-Infograph (see Tab. 9, Appx. A.1.6).

Both observations together suggest that previous domain generalization performance for models trained or evaluated on those datasets needs to be taken with a grain of salt: It is highly likely that their scores are inflated and the models' true OOD generalization capability is lower.

We also analyze the domain composition of datasets from Mayilvahanan et al. (2023), who created several subsets of LAION-200M filtered for samples that are *highly similar* to ImageNet OOD test sets. The removed images are expected to be (near-) duplicates of test images in terms of both content and style. Their dataset *'combined-pruned'* is a subset of LAION-200M where highly similar images to ImageNet-Sketch, ImageNet-R, ImageNet-Val2, ImageNet-Val, ImageNet-A, and ObjectNet were pruned. In their work, it remained unclear whether pruning also effectively removed all images of the rendition domain, which we can now answer.

Tab. 2 reveals that a considerable number of renditions remains in the pruned dataset (at least $6.4\%$, corresponding to around 11 million images). These remaining renditions might have played a significant role in the generalization performance of their CLIP models, especially on ImageNet-Sketch and ImageNet-R. As a result, CLIP's domain generalization performance is yet to be evaluated fairly. We refer the reader to Appx. A.1.6 for further analysis on domain composition at different domain classifier validation-precision levels.

### 3.4 SINGLE-DOMAIN DATASETS

We now use our domain classifiers at $98\%$ validation-precision to subsample LAION-200M. We obtain LAION-Natural with roughly 57 million samples and LAION-Rendition with roughly 16 million samples. Fig. 3 shows random samples from both datasets, more samples are shown in Figs. 20 and 21. We also deploy the domain classifiers on the ImageNet and DomainNet test sets to remove the domain contamination reported above and create *clean test sets*. The exact number of datapoints and the number of classes for each test set are detailed in Tab. 12. These datasets enable us to more fairly assess CLIP's out-of-domain generalization performance in the following sections.

## 4 REFINING THE EVALUATION OF CLIP'S OOD PERFORMANCE

**Training Details** For all our experiments, we train CLIP ViT-B/32 (Dosovitskiy et al., 2020) from scratch for 32 epochs with a batch size of 16 384 on a single node with either four or eight

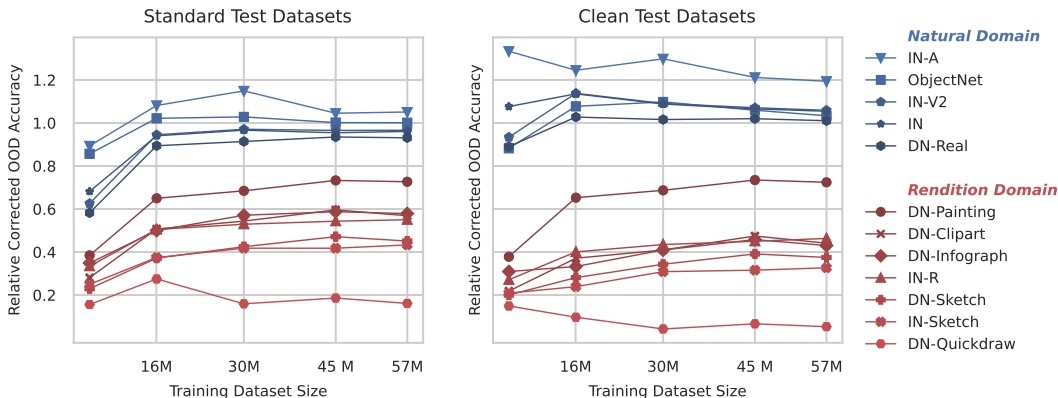

Figure 3: **Random samples from LAION-Natural and LAION-Rendition**.

Figure 4: **Across scales, CLIP performs substantially poorer on unseen domains**. The *relative corrected OOD accuracy* shows performance losses or gains of a CLIP model trained exclusively on the *natural domain* via LAION-Natural compared to a CLIP model trained on an equally-sized subsample of the domain-contaminated LAION-200M. We evaluate models on the standard ImageNet and DomainNet test sets (left) and our cleaned versions of them (right, see Sec. 3.4). When training only on samples from the *natural domain*, we see a decrease in performance for both standard and cleaned test datasets (i.e., relative performance $< 1$). This means that without samples from the *rendition domain*, CLIP's generalization ability suffers significantly and consistently across scales.

A100 GPUs (training takes several days, depending on dataset size). We use the implementation and hyperparameters provided by Ilharco et al. (2021).

We now return to our central question: To what degree is CLIP's ability to generalize to renditions influenced by seeing many renditions during training? To answer this, we first train CLIP on the full 57-million-sample LAION-Natural dataset, as well as random subsets of $45$ million, $30$ million, $16$ million, and $4$ million samples. We then compare the classification accuracy of these models to CLIP models trained on equally sized random subsets of LAION-200M, reporting the accuracy ratio, which we term *relative corrected OOD accuracy*. We evaluate this metric on the original ImageNet and DomainNet test sets as well as on our cleaned versions (see Sec. 3.4). The results are summarized in Fig. 4.

Across the board, we find that the relative corrected OOD accuracy on the clean datasets is around or above $1.0$ for *natural* test sets but drops to around $0.4$ for most *rendition* test sets. This demonstrates that, without domain contamination of the training distribution, CLIP does not generalize across domains nearly as effectively as previously assumed. Notably, the relative corrected OOD accuracy is very consistent across dataset scales, allowing us to conjecture that this result also holds for CLIP models trained on even larger data sizes. For raw accuracy comparisons of LAION-Natural vs. LAION, we refer the reader to Appx. A.2.2.

To further reinforce this observation, we build LAION-Mix-$n$M by replacing $n$ million samples from LAION-Natural with samples from LAION-Rendition. As shown in Tab. 3, replacing $13$ or $16$ million samples with renditions has minimal impact on performance in the *natural* domain but

Table 3: **Performance on the *rendition* domain is largely driven by renditions in the training data**. We compare the top-1 accuracy of CLIP trained *without* renditions on LAION-Natural to CLIP trained on datasets of the same size *with* renditions: LAION-Mix-$n$M contains $n$ million renditions, LAION-Rand is a random subset of LAION-200M with an estimated fraction of 7.9 % – 13.86 % renditions (see Tab. 2). Training with renditions greatly impacts performance on the *rendition* domain. The *natural* column shows the average performance of each model on ImageNet-A, ObjectNet, ImageNet-V2, ImageNet-Val, and DomainNet-Real, while the *rendition* column reflects the average performance on DomainNet-Painting, DomainNet-Clipart, DomainNet-Infograph, DomainNet-Sketch, DomainNet-Quickdraw, ImageNet-R, and ImageNet-Sketch.

| Dataset | Standard Datasets top-1 Acc. | | Clean Datasets top-1 Acc. | |
|---|---|---|---|---|
| | *Natural* | *Rendition* | *Natural* | *Rendition* |
| LAION-Natural | 36.88 % | 21.98 % | 39.72 % | 17.81 % |
| LAION-Mix-13M | 37.28 % | 40.48 % | 38.97 % | 40.78 % |
| LAION-Mix-16M | 36.92 % | 41.46 % | 38.58 % | 42.07 % |
| LAION-Rand-57M | 37.62 % | 40.66 % | 36.99 % | 39.58 % |

significantly boosts performance in the *rendition* domain (near 100 % increase) compared to the model trained solely on LAION-Natural, highlighting the effect of domain contamination. For comparison, we also include the performance of a CLIP model trained on LAION-Rand-57M (57 million random subsample of LAION-200M), which outperforms the LAION-Natural model on rendition domains. This is likely due to LAION-Rand-57M containing an estimated 7.9 % – 13.86 % renditions and a higher proportion of ambiguous samples (25.41 % – 63.70 %).

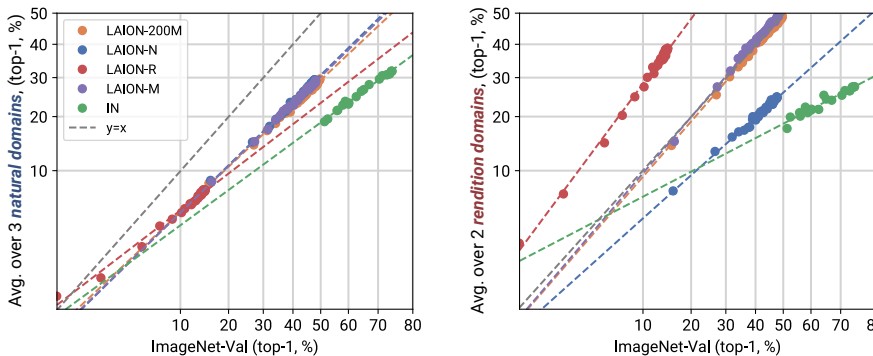

Figure 5: **CLIP's effective robustness to renditions is driven by domain contamination**. We evaluate effective robustness (Fang et al., 2022; Taori et al., 2020) of models trained on different LAION-200M subsets. **Left**: The y-axis represents average accuracy on ImageNet-centric *natural* domain datasets (ImageNet-A, ObjectNet, ImageNet-V2). **Right**: The y-axis shows average performance on ImageNet-centric *rendition* datasets (ImageNet-Sketch, ImageNet-R). **Overall**, CLIP trained on LAION-Natural matches the effective robustness of a LAION-200M-trained CLIP on the *natural* domain but has significantly lower effective robustness on the *rendition* domain. This shows that CLIP requires rendition samples in its training distribution to perform well on this domain.

To put the relative corrected OOD accuracy of Fig. 4 in context, we also evaluate effective robustness on the *natural* and *rendition* domains. Fig. 5 shows the top-1 classification accuracy of multiple CLIP models trained on LAION-200M, LAION-Natural, LAION-Rendition, LAION-Mix, and ResNets trained on ImageNet (see Appx. A.3 for more details on ResNet training). The x-axis shows performance on ImageNet-Val. The y-axis represents average accuracy on ImageNet-centric *natural* domain datasets (ImageNet-A/V2, ObjectNet) for the left plot and average performance on ImageNet-centric *rendition* datasets (ImageNet-Sketch/R) for the right plot. We show results for the 13 million version of LAION-Mix as it aligns closely with the effective robustness of LAION models. As expected, models with the same training regimen align along a line, with the $y$-offset from the ImageNet line indicating effective robustness. While all models trained on LAION subsets

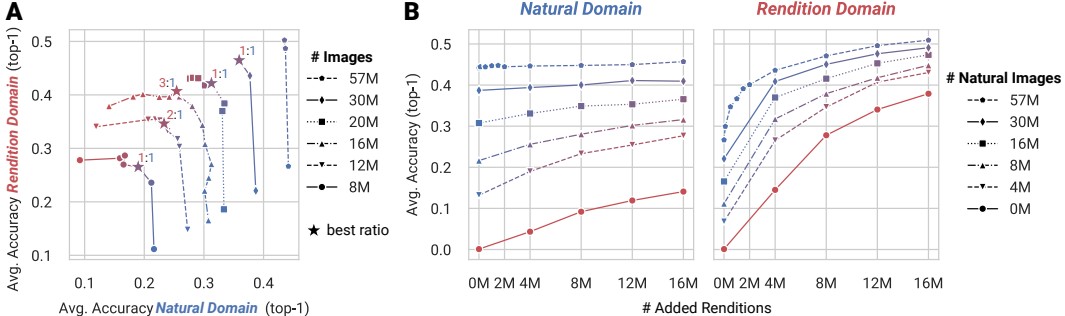

Figure 6: **A**: **Optimal data mixture**. We show the average accuracy on the *natural* and *rendition* domain for models trained with LAION-Mix of different absolute sizes and rendition-to-natural ratios (red indicates only renditions and blue only natural images). The best overall performance (corresponding to the point furthest from the origin) is achieved with a rendition-to-natural ratio between 1:1 and 3:1, which is consistent across scales. **B**: **Effect of adding renditions**. We also analyze model performance with increasingly more renditions added to a fixed-size training set of natural images (which increases overall dataset size). The amount of additional rendition samples required to reach a specific performance on the rendition domain depends on the number of natural samples included in the training set. While natural training samples give some performance boost on the rendition domain, rendition samples do this much more efficiently.

achieve similar effective robustness on the *natural* domain (Fig. 5 left), effective robustness on the *rendition* domain varies greatly and is notably lowest for LAION-Natural-trained models. Effective robustness plots on the individual ImageNet and DomainNet test sets can be found in Appx. A.4. Combining the findings in this section, we now answer our original question: To what extent does domain contamination explain CLIP's performance on renditions?

*Domain contamination contributes substantially to CLIP's strong performance on renditions.*

## 5 INVESTIGATING DOMAIN MIXING AND SCALING EFFECTS

In the previous section, we explored training on single-domain datasets. Equipped with these clean datasets, we can now, for the first time, conduct a controlled investigation on what happens when large-scale datasets from different domains are mixed. First, we show performance on the *natural* and *rendition* domain for models trained on LAION-Mix of different sizes and mixing ratios in Fig. 6A. Varying the mixing ratio while keeping the overall training set size constant reveals that a rendition-to-natural ratio between 1:3 and 1:1 achieves the best overall performance. This optimal range is consistent across training set sizes, although insights on larger scales are limited by the availability of LAION-Rendition samples (in total 16 million images). We hope our results can help practitioners while mixing such domains.

In our second experiment, we progressively add more rendition samples to fixed-size training sets of natural images (Fig. 6B). We find that models starting with more natural images require far fewer renditions to achieve the same performance on the rendition domain. This suggests that large amounts of natural images help the model learn some features that can be useful for generalizing to renditions, and relatively few additional renditions suffice to reach good performance on the rendition domain. In addition to boosting the performance on rendition test sets, adding rendition samples to the training set marginally boosts the performance on natural test sets, albeit with quickly diminishing returns. While performance in the natural domain benefits from rendition samples, natural samples are much more helpful. Likewise, training on few rendition samples gives higher performance than training on substantially more natural samples (see Fig. 6B, Tab. 3)—echoing our conclusion in Sec. 4 that CLIP does slightly generalize but much less than previously assumed.

## 6 DISCUSSION

**Contextualizing our core result**   The literature often assumes that CLIP is capable of generalizing OOD (Radford et al., 2021; Abbasi et al., 2024; Nguyen et al., 2024; Fang et al., 2022; Li et al., 2023; Shu et al., 2023). Our main result is that CLIP's strong generalization to rendition domains is largely due to the presence of samples from those domains in its training distribution. Fang et al. (2022) showed CLIP's robustness is tied to its data distribution but do not mention any specific characteristic. In contrast, Mayilvahanan et al. (2023) indicate that other dataset properties, not train-test similarity on a per-sample level, influence robustness. We conclusively demonstrate that CLIP's apparent OOD robustness on standard OOD benchmarks like ImageNet-Sketch or ImageNet-R is often an artifact of overlapping domain data, rather than genuine OOD generalization. This refines the conclusion of Fang et al. (2022) and directly challenges Mayilvahanan et al. (2023) (see Sec. 3.3 and Sec. 4), and several other works (Radford et al., 2021; Abbasi et al., 2024; Nguyen et al., 2024; Fang et al., 2022; Li et al., 2023; Shu et al., 2023). To the best of our knowledge, no work exists that addresses OOD generalization without domain contamination at this paper's scale (10s of millions).

**Validity of conclusions for larger datasets**   Although our training sets are constrained by the availability of natural and rendition samples, we believe that the insights gained from analyzing datasets with sizes spanning over one order of magnitude will remain applicable to even larger datasets. Specifically, the disparity in 'relative corrected accuracy' shown in Fig. 4 remains stable across dataset sizes from 4M to 57M. Similarly, effective robustness illustrated in Fig. 5 is influenced by the training distribution rather than the dataset size, which is also supported by findings in previous works (Miller et al., 2021; Fang et al., 2022; Mayilvahanan et al., 2023). Lastly, CLIP's performance scales predictably across domain mixtures as shown in Sec. 5. Overall, we see no indication that our results should not transfer to larger scales.

**Validity of conclusions for other architectures and loss functions**   Prior work strongly supports the generalizability of our findings on data contamination and optimal ratios across architectures and training methods beyond CLIP (Miller et al., 2021; Fang et al., 2022). For instance, Fang et al. (2022) demonstrates that CLIP's robustness is driven primarily by the training distribution, with factors like dataset size, language supervision, and contrastive loss playing minimal roles. They also show that models trained on identical data distributions, regardless of loss functions (e.g., SimCLR+FT, CLIP, Supervised) or architectures (e.g., varying backbones and parameter sizes), exhibit similar effective robustness. This indicates that our conclusions are likely to hold across model types. We further address their validity across dataset sizes in Sec. 6.

**Choice of domain and validity of conclusions for other domains**   For models to align with human perception, it is essential that they generalize to rendition domains, particularly in out-of-distribution (OOD) scenarios. Humans are adept at interpreting abstract visual renditions (Hendrycks et al., 2021a), while machines often depend primarily on textural cues (Geirhos et al., 2019). Consequently, we focus on natural images vs. renditions as our subject of study. Our methodology can be applied to evaluate OOD generalization for other domains, and we expect that our findings will hold true, as domain contamination is a general problem not tied to the specific domains we examined. However, we do anticipate challenges in accurately characterizing certain domain shifts, which could impede training the domain classifier. Nonetheless, if a small labeled dataset can be created to differentiate between these shifts, the subsequent processes should proceed smoothly. Given the manual effort required and the potential redundancy in findings, we defer this task to future work.

## 7 CONCLUSION

With the emergence of models trained on web-scale datasets containing abundant samples from seemingly all possible domains, the study of domain generalization mostly came to a halt. Hence, the question of how dataset scale actually affects the ability of models to generalize between domains remains unanswered. Here, we try to answer this question thoroughly by fully controlling the domains used for model training. By creating clean subsets of LAION containing either natural images or renditions, and by training models on various mixtures and dataset sizes, we show that the generalization performance of CLIP trained on only one domain drops to levels similar to what has been observed for ImageNet-trained models. Hence, we conclude that the domain generalization problem remains unsolved even for very large-scale datasets. We release all training set splits as well as pretrained models and encourage the field to re-consider domain generalization as a central benchmark for future progress on model architectures, inductive biases, and learning objectives.

REPRODUCIBILITY

We describe the methodology to create all of the datasets we use in Secs. 3.1 and 3.4 and Appx. A.1.1. We also detail our domain classifiers and their training in Sec. 3.2 and Appx. A.1.2 and A.1.3. Further, the training details of the CLIP models and the ResNet models are in Sec. 4 and Appx. A.3. This should be sufficient to reproduce all our experimental results. We will release all of our labeled datasets, all cleaned test datasets, our LAION-400M subsets (LAION-Natural and LAION-Rendition), the domain classifiers, and the CLIP model checkpoints. All of these resources are already uploaded to HuggingFace and will be made public at acceptance of this paper. The source code for all experiments can be found in the supplementary material and will be publicly released, too. For training the CLIP models we used the publicly available code from (Ilharco et al., 2021) exclusively.

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

# A APPENDIX

## A.1 MORE DETAILS ON THE DOMAIN CLASSIFIER

### A.1.1 LABELING

As mentioned in Sec. 3.1, we take a *texture-centric* approach in domain labeling. We resolve further ambiguities with respect to labeling in the following way:

- Natural objects with watermark or text, infographs with natural objects, signs with human symbol (eg. walking signal), objects with common logos (eg. Nike), naturalistic books or movie covers, images that are retro / low resolution / blurry / grainy / or with fake background but with texture information preserved, graphically altered natural images with significant texture information, and real objects with fake backgrounds **are all classified as natural**.

- Stylistic: Infographs with stylized objects, stylized books or movie covers, retro / low resolution / blurry / grainy /graphically altered images with significant loss in texture information, stylized objects on plain or common natural background (eg. wall, bedsheet etc.) **are all classified as stylistic**.

- Ambiguous: Tattoos where hand / back is very visible, sculpture with real objects around, real images with distinct drawing of logos with objects, images that are retro / low resolution / blurry / grainy / or with fake background but with little texture information preserved **are all classified as ambiguous**.

The labeling of 19,000 images were done by one annotator who labeled about 750-1000 images per hour. The annotator also did a checking of these labels by regrouping and going over them again. Two other annotators re-labeled the test set, a collection of 3000 images to affirm the quality of labels and the domain classifier (see Appx. A.1.4). All annotators are the authors of the work. We visualize our labeling setup in Fig. 7. We also state the final breakdown of labeled images in Tab. 4.

Table 4: **Number of labeled data points from several datasets and their domain-wise breakdown**. For training our domain classifier, we use the LAION-200M (Train), and LAION-200M (Val) for validation, and everything else to evaluate the final test performance.

| Dataset | Natural | Stylistic | Ambiguous | Total |
|---|---|---|---|---|
| LAION-200M (Train) | 7268 | 2978 | 2754 | 13000 |
| LAION-200M (Val) | 1000 | 1000 | 1000 | 3000 |
| LAION-200M (Test) | 1000 | 1000 | 1000 | 3000 |
| ImageNet-A | 974 | 7 | 19 | 1000 |
| ObjectNet | 917 | 2 | 81 | 1000 |
| ImageNet-R | 22 | 859 | 119 | 1000 |
| ImageNet-Sketch | 49 | 937 | 14 | 1000 |
| ImageNet-V2 | 945 | 5 | 50 | 1000 |
| ImageNet-Val | 934 | 16 | 50 | 1000 |
| DomainNet-Clipart | 48 | 933 | 19 | 1000 |
| DomainNet-Infograph | 134 | 720 | 146 | 1000 |
| DomainNet-Painting | 101 | 795 | 104 | 1000 |
| DomainNet-Quickdraw | 0 | 1000 | 0 | 1000 |
| DomainNet-Real | 836 | 111 | 53 | 1000 |
| DomainNet-Sketch | 24 | 942 | 34 | 1000 |

## A.1.2 OTHER METHODS FOR TRAINING DOMAIN CLASSIFIERS

Apart from the domain classifier training methods explored in Sec. 3.2, we explore a few more as follows:

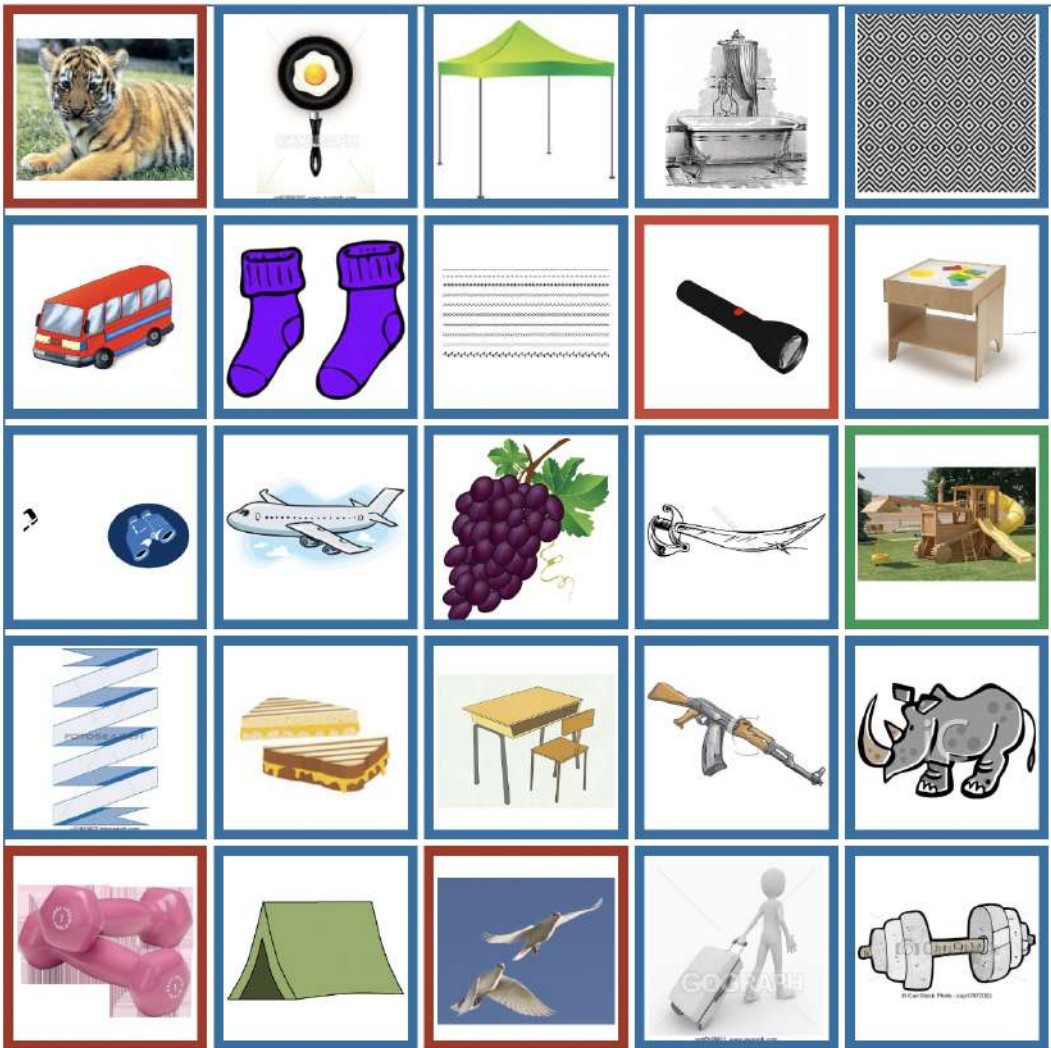

Figure 7: **Labeling setup.** By clicking on the image, the border changes to red, green, or blue, each representing natural, ambiguous, or rendition. By pressing the right or the left button the previous or next set of 25 images are rendered and the labels of the previous images are updated in a json file.

**Contrastive Style Descriptors (CSD)** Somepalli et al. (2024) fine-tune pre-trained backbones via multi-label supervised contrastive learning and self-supervised learning with only style-preserving augmentations (random flips, resize, rotation). The resulting final-layer embeddings serve as style descriptors: During inference, they find the $k$ stylistically nearest neighbors in a database of labeled images (e.g., the training set) by computing pairwise embedding-similarities to the test images. An image is classified as belonging to a style if at least one of the $k$ neighbors has that style. We can directly set up their method using the $13\,000$ labeled LAION-200M images as both the training set and the database for inference. From that, we obtain two binary classifiers, CSD-N (classifying natural vs. non-natural) and CSD-R (classifying renditions vs. non-renditions), which jointly can be used for our ternary classification.

**Centroid Embeddings** Inspired by the baselines used by Somepalli et al. (2024), we implement a simple model (embedding model plus linear readout). Here, we take the pre-trained CLIP ViT-L/14 as the embedding model and create a linear readout by comparing embeddings to the centroid embedding of each domain. We use this as a ternary untrained nearest-neighbor classifier, dubbed CE.

Table 5: **We chose the best *natural* classfier and the best *rendition* classifier** amongst binary classifiers based on Contrastive Style Descriptors (CSD) (Somepalli et al., 2024) and Density Ratios (DR) (Cohen-Wang et al., 2024b) as well as ternary classifiers using a linear readout based on either each domain's centroid embedding (CE) or a fine-tuned CLIP (FT). All models use CLIP ViT-L/14 pretrained on LAION-2B. We report precision and recall on for the *natural* class (top) and *rendition* class (bottom) on ImageNet (IN) and DomainNet (DN) test sets and average performance across all test sets. Model hyperparameters are chosen for a validation precision of $98\%$ if possible. For each class, we select the classifier with the highest recall on the validation.

| cls=natural | Val | | Test | | IN-Val | | IN-v2 | | IN-A | | ON | | DN-R | | Average | |
|---|---|---|---|---|---|---|---|---|---|---|---|---|---|---|---|---|
| **Model** | P | R | P | R | P | R | P | R | P | R | P | R | P | R | P | R |
| CSD-N k=1 | 0.61 | 0.85 | 0.58 | 0.85 | 0.96 | 0.93 | 0.97 | 0.92 | 0.98 | 0.91 | 0.93 | 0.94 | 0.92 | 0.88 | 0.85 | 0.90 |
| CSD-R k=23 | 0.98 | 0.26 | 0.99 | 0.29 | 1.00 | 0.22 | 1.00 | 0.27 | 1.00 | 0.27 | 1.00 | 0.59 | 0.99 | 0.32 | 0.99 | 0.32 |
| DR-N | 0.98 | 0.00 | 0.50 | 0.00 | 0.00 | 0.00 | 0.00 | 0.00 | 0.00 | 0.00 | 0.00 | 0.00 | 0.00 | 0.00 | 0.21 | 0.00 |
| DR-R | 0.98 | 0.08 | 0.72 | 0.08 | 1.00 | 0.00 | 1.00 | 0.00 | 1.00 | 0.00 | 0.95 | 0.20 | 1.00 | 0.00 | 0.95 | 0.05 |
| CE | 0.98 | 0.35 | 0.89 | 0.33 | 0.95 | 0.02 | 1.00 | 0.04 | 1.00 | 0.02 | 0.99 | 0.16 | 0.99 | 0.11 | 0.97 | 0.15 |
| FT | 0.98 | 0.41 | 0.95 | 0.44 | 1.00 | 0.36 | 0.99 | 0.40 | 1.00 | 0.46 | 0.99 | 0.53 | 1.00 | 0.42 | 0.99 | 0.43 |

| cls=rendition | Val | | Test | | IN-R | | IN-S | | DN-S | | DN-Q | | DN-P | | DN-C | | DN-I | | Average | |
|---|---|---|---|---|---|---|---|---|---|---|---|---|---|---|---|---|---|---|---|---|
| **Model** | P | R | P | R | P | R | P | R | P | R | P | R | P | R | P | R | P | R | P | R |
| CSD-N k=6 | 0.98 | 0.26 | 0.99 | 0.24 | 1.00 | 0.20 | 1.00 | 0.18 | 1.00 | 0.25 | 0.00 | 0.00 | 1.00 | 0.24 | 1.00 | 0.22 | 0.98 | 0.34 | 0.88 | 0.21 |
| CSD-R k=1 | 0.64 | 0.56 | 0.68 | 0.60 | 0.93 | 0.62 | 0.98 | 0.63 | 0.98 | 0.62 | 0.00 | 0.00 | 0.92 | 0.59 | 0.98 | 0.63 | 0.82 | 0.46 | 0.77 | 0.52 |
| DR-N | 0.98 | 0.20 | 0.98 | 0.23 | 1.00 | 0.29 | 1.00 | 0.20 | 1.00 | 0.27 | 1.00 | 0.01 | 1.00 | 0.28 | 1.00 | 0.28 | 0.98 | 0.11 | 0.99 | 0.21 |
| DR-R | 0.98 | 0.35 | 0.98 | 0.41 | 1.00 | 0.60 | 1.00 | 0.71 | 1.00 | 0.74 | 1.00 | 0.33 | 0.99 | 0.60 | 1.00 | 0.65 | 0.98 | 0.39 | 0.99 | 0.53 |
| CE | 0.98 | 0.11 | 0.99 | 0.12 | 0.99 | 0.43 | 1.00 | 0.39 | 1.00 | 0.30 | 1.00 | 0.09 | 0.98 | 0.47 | 1.00 | 0.38 | 1.00 | 0.01 | 0.99 | 0.26 |
| FT | 0.98 | 0.27 | 0.95 | 0.26 | 1.00 | 0.38 | 1.00 | 0.57 | 1.00 | 0.61 | 1.00 | 0.68 | 1.00 | 0.21 | 1.00 | 0.50 | 1.00 | 0.30 | 0.99 | 0.42 |

We use the validation set to determine the two best domain classifiers, one for natural images and one for renditions. Since the domain classifier should maximize precision above all else, we set the confidence threshold for each model such that it achieves $98\%$ per-class precision. For CSD, we instead choose $k$ to reach this precision. Tab. 5 reports each model's precision and recall on the *natural* and *rendition* class across ImageNet and DomainNet test sets. For raw accuracy numbers of all models, which in general are high for most, please refer to Tabs. 7 and 8 in Appx. A.1.5.

### A.1.3 TRAINING DETAILS FOR THE DOMAIN CLASSIFIERS

As mentioned in Sec. 3.2, we train several domain classifiers with several different training procedures. For the baselines (Cohen-Wang et al., 2024b; Somepalli et al., 2024), we simply use the training code detailed in their works and their public code. For the FT (Finetuning) model, as mentioned in Sec. 3.2, we finetune a CLIP ViT-L/14 pretrained on LAION-2B with a linear readout. We finetune all models on 4 A100 GPUs, using a batch size of 256, weight decay of $5e-4$, using an SGD optimizer, with step scheduler (0.1 every 20 epochs), at a learning rate of 0.1, for 50 epochs. All models converge. Each model took about 2 A100 GPU hours to train, therefore all the models took around 30 A100 GPU hours. The storage requirement for these datasets were less than 100 GB memory.

We train these models on the 13K LAION domain dataset or subsets of it with 2 or 3 classes. To compare with the models from Cohen-Wang et al. (2024b), we train binary classifiers where we club natural with ambiguous and differentiate it from rendition (we name this FT-R), or we club rendition with ambiguous and differentiate it from natural (we name this FT-N). Further, we create several subsets for each of the ternary and the binary classification problem by balancing the number of datapoints in each class. We add the prefix '(balanced)' to these models.

### A.1.4 AFFIRMING THE QUALITY OF LABELS AND THE DOMAIN CLASSIFIER

Our primary goal is to create clean versions of natural and rendition datasets. To achieve this, we use domain classifiers at a threshold where the validation set precision is high, ensuring the selection of images that are distinctly 'natural-like' or 'rendition-like'. This allows us to train the domain classifiers with some label noise as long as the most obvious images are correctly classified. Our experimental results (see Sec. 4; Fig. 1, 4, 5; Tab. 3) and visualizations of random samples from

Table 6: **Domain classifiers precision and recall for original and adjusted test set on the corresponding *natural* or *rendition* classes**. For the adjusted test set, two additional annotators labeled each image, and the final label was assigned based on majority agreement, with ambiguous cases labeled as such. We observe no substantial change in precision and recall values indicating the robustness of our pipeline.

| cls=*nat*,*rend* | Test | | Test (Adjusted) | |
|---|---|---|---|---|
| **Model** | **P** | **R** | **P** | **R** |
| FT | 0.95 | 0.45 | 0.95 | 0.48 |
| DR-R | 0.98 | 0.41 | 0.97 | 0.41 |

natural and rendition datasets (see Fig. 3, 21, 22) confirm the reliability of our labeling procedure and our domain classifiers.

Nonetheless, we re-labeled our test set of 3,000 images with two additional independent annotators. We generated new labels for the test set based on the majority vote, labeling images as ambiguous if there was no consensus. We note that the majority vote agreed with our previous labels on 93% of the images. Testing our domain classifiers at a 98% validation precision on this new test set, we found that precision and recall remained high, indicating strong agreement on the clearly 'natural-like' or 'rendition-like' images (see Tab. 6). This further reinforces the overall confidence in the labeling procedure and the domain classifiers.

### A.1.5 RAW DOMAIN CLASSIFIER PERFORMANCE ON LABELED SETS

In the main text in Sec. 3.2 we only compute the precision and recall obtained from the threshold at which we get 98% precision on LAION-200M Val domain dataset. We here report the accuracy of these classifiers on these test sets at their own standard precision of these models. We also train additional classifiers binary and ternary classifiers and by balancing the dataset sizes. The results can be found in Tabs. 7 and 8.

Table 7: **Accuracy on each of the natural test sets on class natural without thresholding.** Some classifiers give the illusion of being good but have very low precision or recall(see Sec. 3.2).

| Model | (Val) | (Test) | IN-Val | IN-V2 | IN-A | ON | DN-R | DN-I |
|---|---|---|---|---|---|---|---|---|
| FT | 0.90 | 0.89 | 0.93 | 0.94 | 0.96 | 0.95 | 0.94 | 0.72 |
| CE | 0.75 | 0.78 | 0.80 | 0.84 | 0.86 | 0.95 | 0.81 | 0.19 |
| FT-N | 0.89 | 0.90 | 0.94 | 0.95 | 0.97 | 0.97 | 0.93 | 0.49 |
| DR-N (balanced) | 0.89 | 0.91 | 0.94 | 0.94 | 0.95 | 0.98 | 0.92 | 0.50 |
| DR-R | 0.98 | 0.97 | 0.99 | 0.99 | 1.00 | 1.00 | 0.97 | 0.90 |
| FT (balanced) | 0.78 | 0.82 | 0.84 | 0.86 | 0.86 | 0.88 | 0.83 | 0.46 |
| FT-R | 0.96 | 0.95 | 0.93 | 0.95 | 0.97 | 0.98 | 0.96 | 0.90 |
| FT-N (balanced) | 0.85 | 0.85 | 0.92 | 0.95 | 0.96 | 0.95 | 0.91 | 0.43 |
| DR-R (balanced) | 0.93 | 0.92 | 0.93 | 0.94 | 0.95 | 0.99 | 0.90 | 0.75 |
| FT-R (balanced) | 0.86 | 0.86 | 0.88 | 0.88 | 0.90 | 0.89 | 0.88 | 0.84 |
| DR-N | 0.93 | 0.92 | 0.94 | 0.95 | 0.94 | 0.99 | 0.92 | 0.76 |

### A.1.6 DOMAIN COMPOSITION AT DIFFERENT PRECISION LEVELS

We provide a detailed overview over the domain composition of datasets at standard precision in Tab. 9, and over the domain composition of datasets at 98% precision in Tab. 10. In Fig. 8, we examine LAION's composition at different validation precision levels. Starting with a lower validation precision threshold (0.33) where both natural and rendition images are present, we observe that the number of ambiguous examples increases at both high and low precision levels, which is expected given that our final domain classification relies on the agreement of two classifiers. Fig. 8 further supports our choice of a 0.98 precision threshold, as it strikes a good balance between precision and the ability to select sufficiently large datasets in the tens of millions.

Table 8: **Accuracy on each of the rendition test sets on class natural without thresholding.** Some classifiers give the illusion of being good but have very low precision or recall(see Sec. 3.2).

| Model | (Val) | (Test) | IN-R | IN-S | DN-S | DN-Q | DN-P | DN-C | DN-I |
|---|---|---|---|---|---|---|---|---|---|
| DR-R | 0.77 | 0.80 | 0.93 | 0.98 | 0.98 | 0.96 | 0.92 | 0.93 | 0.88 |
| FT (balanced) | 0.78 | 0.88 | 0.82 | 0.94 | 0.94 | 0.91 | 0.80 | 0.85 | 0.77 |
| FT | 0.76 | 0.75 | 0.75 | 0.91 | 0.90 | 0.95 | 0.73 | 0.80 | 0.74 |
| DR-N | 0.89 | 0.92 | 0.99 | 0.99 | 0.99 | 0.98 | 0.97 | 0.97 | 0.94 |
| FT-R | 0.69 | 0.68 | 0.69 | 0.81 | 0.80 | 0.79 | 0.65 | 0.72 | 0.67 |
| DR-N (balanced) | 0.93 | 0.94 | 0.97 | 0.99 | 0.99 | 1.00 | 0.95 | 0.94 | 0.99 |
| FT-R (balanced) | 0.86 | 0.84 | 0.80 | 0.92 | 0.91 | 0.90 | 0.75 | 0.83 | 0.88 |
| CE | 0.61 | 0.62 | 0.95 | 0.90 | 0.89 | 0.96 | 0.95 | 0.93 | 0.32 |
| DR-R (balanced) | 0.90 | 0.93 | 0.99 | 0.99 | 0.99 | 0.99 | 0.98 | 0.97 | 0.96 |
| FT-N | 0.84 | 0.83 | 0.72 | 0.83 | 0.82 | 0.48 | 0.63 | 0.77 | 0.97 |
| FT-N (balanced) | 0.87 | 0.86 | 0.75 | 0.93 | 0.91 | 0.96 | 0.64 | 0.88 | 0.98 |

Table 9: **Domain composition of datasets at standard precision (without thresholding).** The first three columns show the fraction of samples in the original dataset classified as natural, stylistic, or ambiguous, respectively, while the latter column shows the dataset's total number of samples.

| Dataset | Natural [%] | Stylistic [%] | Ambiguous [%] | Total |
|---|---|---|---|---|
| LAION-200M | 60.74 | 13.86 | 25.41 | 199 663 250 |
| ImageNet (Train) | 89.2 | 1.18 | 9.62 | 1 281 167 |
| ImageNet (Val) | 89.1 | 1.18 | 9.72 | 50 000 |
| ObjectNet | 90.22 | 0.1 | 9.68 | 18 574 |
| ImageNet-V2 | 88.49 | 1.38 | 10.13 | 10000 |
| ImageNet-A | 93.79 | 0.52 | 5.69 | 7 500 |
| ImageNet-R | 9.75 | 64.42 | 25.83 | 30 000 |
| ImageNet-Sketch | 3.69 | 85.34 | 10.97 | 50 889 |
| DomainNet-Real | 80.07 | 7.59 | 12.34 | 175 327 |
| DomainNet-Quickdraw | 1.35 | 93.27 | 5.38 | 172 500 |
| DomainNet-Clipart | 8.28 | 75.89 | 15.83 | 48 833 |
| DomainNet-Painting | 13.97 | 56.33 | 29.7 | 75 759 |
| DomainNet-Sketch | 3.1 | 84.18 | 12.71 | 70 386 |
| DomainNet-Infograph | 11.17 | 53.41 | 35.41 | 53 201 |

### A.1.7 ON THE DOMAIN COMPOSITION OF MAYILVAHANAN ET AL. (2023)

Please find in Tab. 11 the exact number of rendition examples calculated by deploying our domain classifier on each the 3 datasets (pruned using rendition test sets) from Mayilvahanan et al. (2023). We see that at least 11-13M images are not pruned away from the datasets, therefore explaining the insignificant drop in performance.

### A.1.8 PREPARING CLEAN DATASETS

In Sec. 3.4, we created several train and test sets from LAION-200M and ImageNet / DomainNet shifts respectively, by deploying our classifier at 98% precision. The exact number of samples and the number of (remaining) classes are in Tab. 12.

Table 10: **Domain composition of datasets at 98% precision.** The first three columns show the fraction of samples in the original dataset classified as natural, stylistic, or ambiguous, respectively, while the latter column shows the dataset's total number of samples.

| Dataset | Natural [%] | Stylistic [%] | Ambiguous [%] | Total |
|---|---|---|---|---|
| LAION-200M | 28.4 | 7.9 | 63.7 | 199 663 250 |
| ImageNet (Train) | 36.0 | 0.4 | 63.6 | 1 281 167 |
| ImageNet (Val) | 35.73 | 0.37 | 63.9 | 50 000 |
| ObjectNet | 50.32 | 0.0 | 49.68 | 18 574 |
| ImageNet-V2 | 36.04 | 0.29 | 63.67 | 10000 |
| ImageNet-A | 43.25 | 0.16 | 56.59 | 7 500 |
| ImageNet-R | 3.56 | 52.82 | 43.61 | 30 000 |
| ImageNet-Sketch | 1.21 | 67.92 | 30.87 | 50 889 |
| DomainNet-Real | 34.31 | 3.98 | 61.71 | 175 327 |
| DomainNet-Quickdraw | 0.09 | 34.41 | 65.5 | 172 500 |
| DomainNet-Clipart | 3.46 | 62.53 | 34.01 | 48 833 |
| DomainNet-Painting | 5.3 | 47.55 | 47.15 | 75 759 |
| DomainNet-Sketch | 1.38 | 69.58 | 29.04 | 70 386 |
| DomainNet-Infograph | 1.59 | 28.11 | 70.3 | 53 201 |

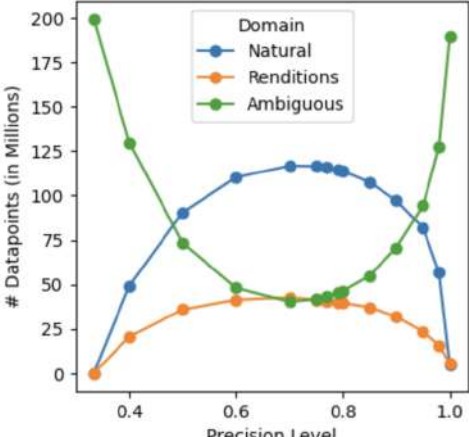

Figure 8: **Domain composition of LAION-200M at different precision levels.** We see the evolution of domain composition of the LAION-200M dataset, determined using the domain classifiers at various precision levles from the validation set.

Table 11: **Number datapoints within the dataset vs number of datapoints pruned away in Mayilvahanan et al. (2023).**

| Dataset | Size | Within | Pruned |
|---|---|---|---|
| sketch-pruned | 191 481 491 | 24 016 047 | 3 654 180 |
| r-pruned | 194 088 525 | 24 304 991 | 3 365 236 |
| combined-pruned | 187 471 515 | 22 173 006 | 5 497 221 |
| sketch-pruned (98% precision) | 19 1481 491 | 13 266 999 | 2 482 751 |
| r-pruned (98% precision) | 194 088 525 | 13 338 759 | 2 410 991 |
| combined-pruned (98% precision) | 187 471 515 | 11 999 276 | 3 750 474 |

Table 12: **Clean datasets composition.** Obtained by deploying the domain classifiers from Sec. 3.2 at 98% precision.

| Dataset | Classes | Size |
|---------|--------:|-----:|
| LAION-Natural | - | 56 685 759 |
| LAION-Stylistic | - | 15 749 750 |
| ImageNet-Val | 985 | 17 864 |
| ImageNet-V2 | 926 | 3 604 |
| ImageNet-Sketch | 991 | 34 564 |
| ImageNet-R | 200 | 15 847 |
| ImageNet-A | 197 | 3 244 |
| ObjectNet | 113 | 9 347 |
| DomainNet-Real | 339 | 60 148 |
| DomainNet-Quickdraw | 344 | 59 353 |
| DomainNet-Infograph | 345 | 14 957 |
| DomainNet-Clipart | 345 | 30 536 |
| DomainNet-Sketch | 344 | 48 974 |
| DomainNet-Painting | 345 | 36 020 |

## A.2 NOTES ON THE CLIP MODELS

### A.2.1 RESOURCES SPENT

We train about 28 CLIP ViT-B/32 models on several subsets of LAION-200M. These models took about 8000 A100 GPU hours. We also needed about 18 TB of memory to store these datasets.

### A.2.2 RAW ACCURACY NUMBERS OF CLIP TRAINED ON LAION-N VS LAION

In Sec. 4, in Fig. 4, we only reported the relative numbers. Here, in Fig. 9, 11, 10, 12, we report the actual numbers as a function of dataset size.

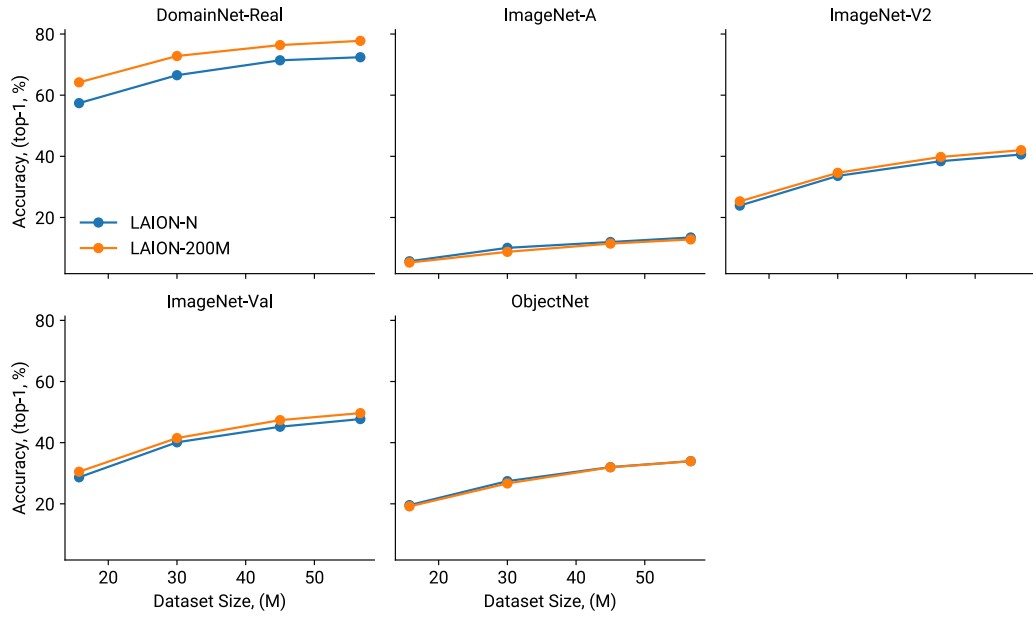

Figure 9: **CLIP trained on LAION v LAION-N performance on standard natural test sets.**

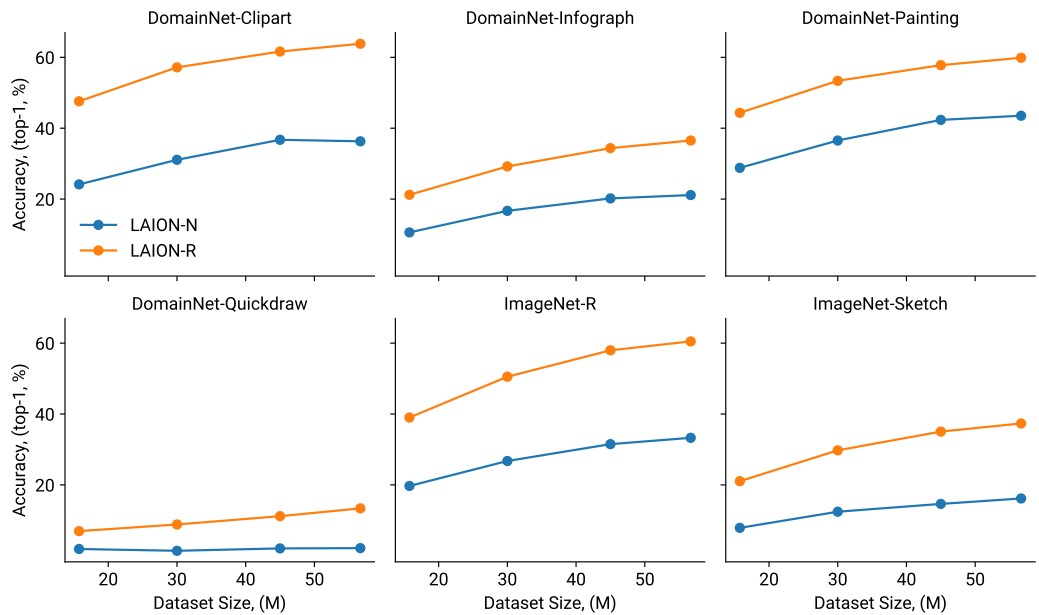

Figure 10: **CLIP trained on LAION v LAION-N performance on standard rendition test sets.**

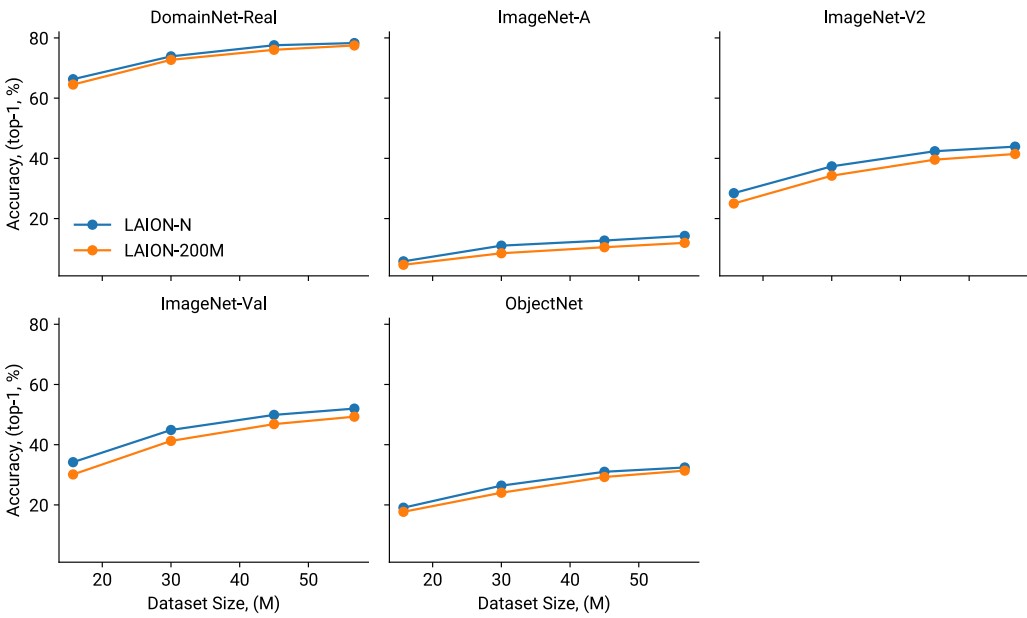

Figure 11: **CLIP trained on LAION v LAION-N performance on clean natural test sets.**

### A.3 TRAINING RESNETS ON IMAGENET

We deploy our natural domain classifier from Sec. 3 at 90% precision (threshold obtain from LAION 13K Val set) on ImageNet-Train to obtain about 1M datapoints belonging to the natural domain (dubbed ImageNet-N). We create several datasets of smaller sizes subsampling from ImageNet-N. We also create randomly sampled datasets of similar sizes from the original ImageNet. We train ResNet-50 models on all of these datasets. We follow the training recipe A3 of Wightman et al. (2021) and train the models for 200 epochs. We then evaluate these models on standard test sets and

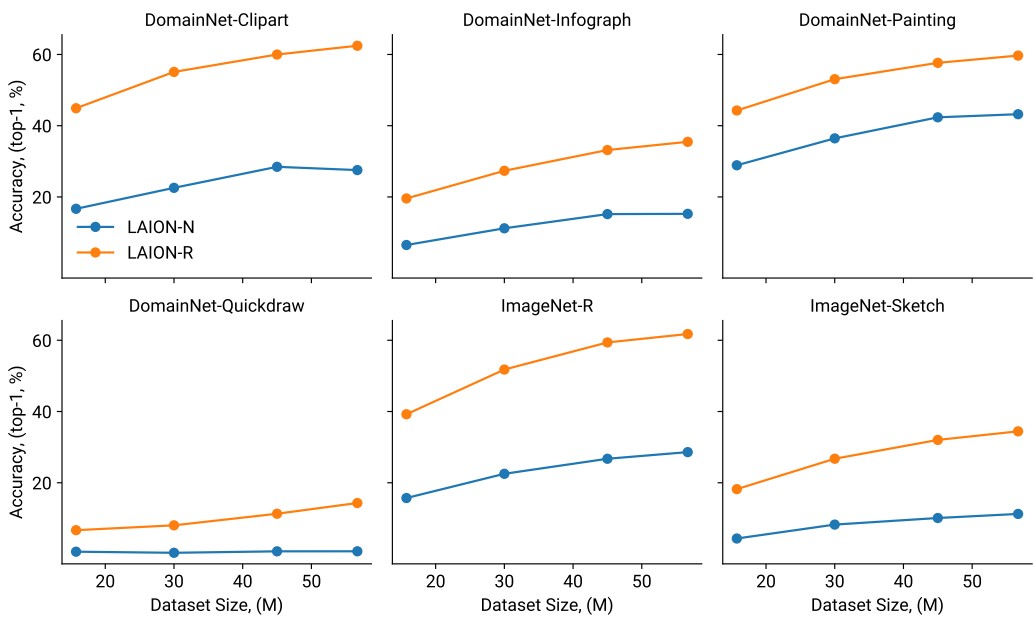

Figure 12: **CLIP trained on LAION v LAION-N performance on clean rendition test sets.**

clean test sets from Sec. 3.4. The accuracies of ResNets trained on subsets of original ImageNet is used for the effective robustness plots in Sec. 4, A.4. Further, the comparison of accuracies between the models trained on subsets from ImageNet-N and ImageNet is in Fig. 13, 15, 14, 16. As such there is no significant performance difference anywhere, thus indicating that ImageNet does not have substantial domain leakage.

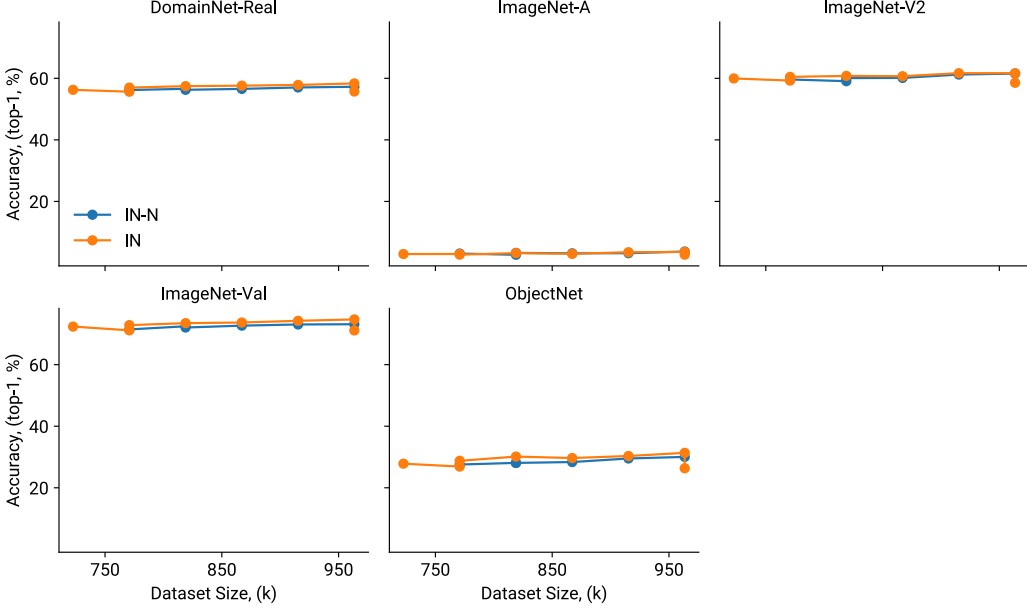

Figure 13: **Resnets trained on ImageNet v ImageNet-N performance on standard natural test sets.**

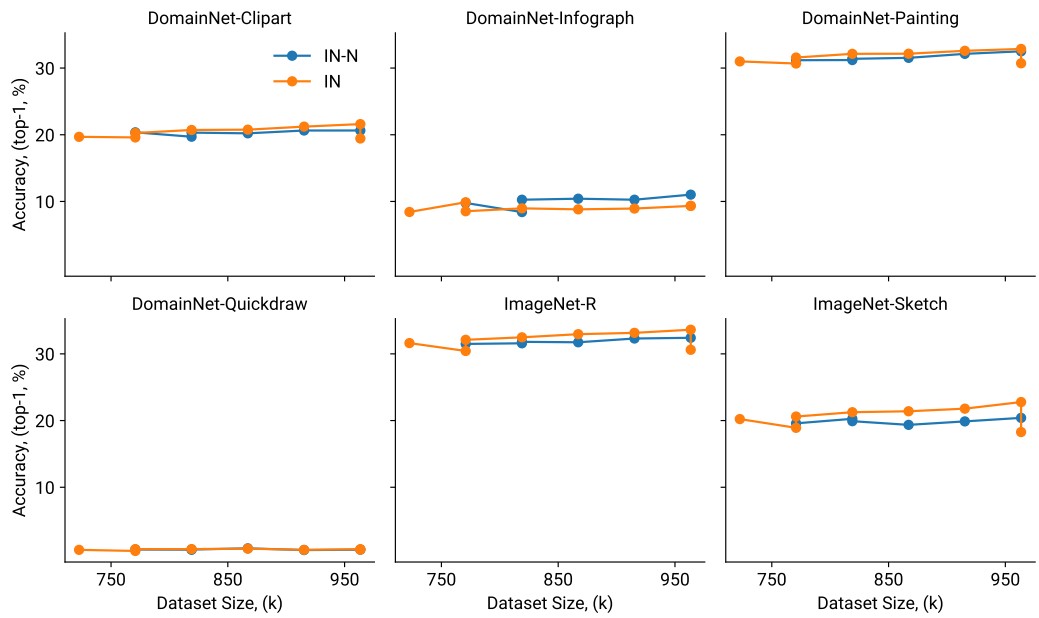

Figure 14: **Resnets trained on ImageNet v ImageNet-N performance on standard rendition test sets.**

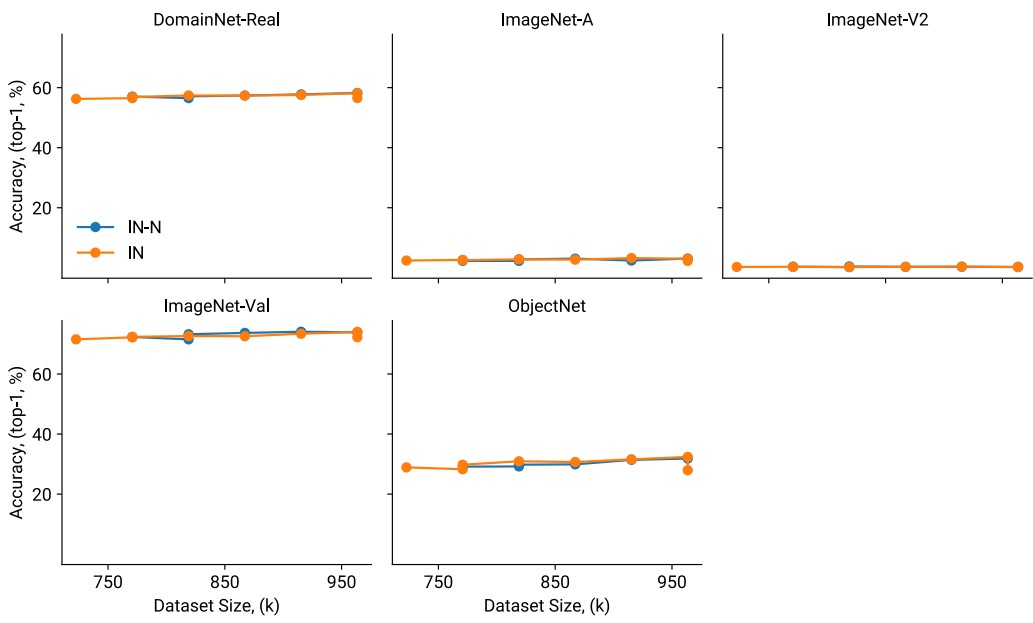

Figure 15: **Resnets trained on ImageNet v ImageNet-N performance on clean natural test sets.**

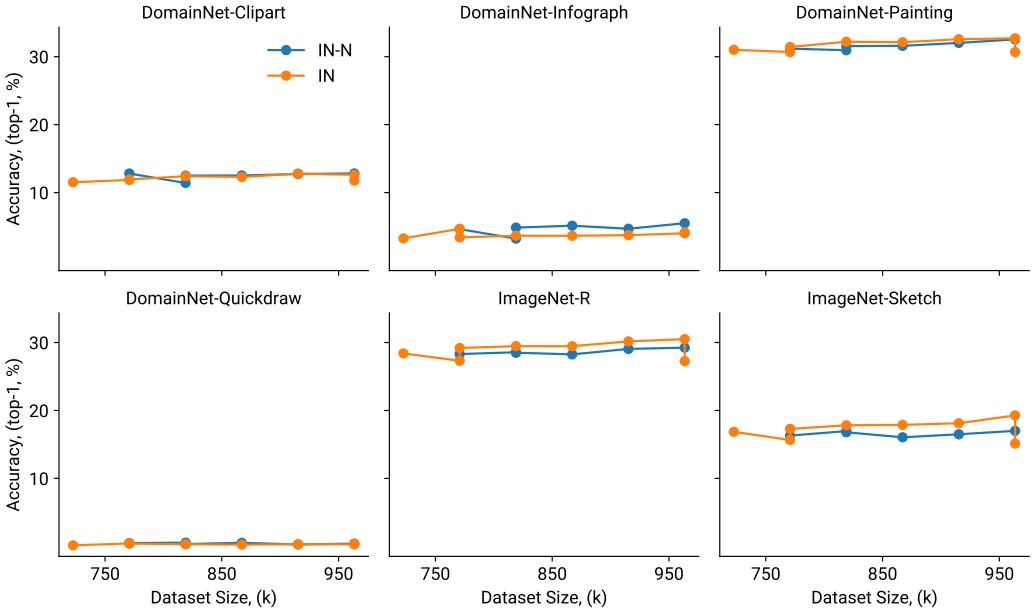

Figure 16: **Resnets trained on ImageNet v ImageNet-N performance on clean rendition test sets.**

## A.4 DETAILED EFFECTIVE ROBUSTNESS PLOTS ON INDIVIDUAL SHIFTS

In Fig. 5 in the main manuscript, we report aggregated results where we average over natural and stylistic ImageNet distribution shifts. We display the results on the individual distribution shifts in Fig. 17. On ImageNet-R and ImageNet-Sketch (bottom row), we observe that the effective robustness of the CLIP models can be modulated by training it on the different dataset splits, i.e. LAION-Natural, LAION-Rendition, LAION-Mix. The model trained on LAION-Natural is much closer to the ImageNet trained model in terms of effective robustness compared to the model trained on LAION-Rendition. In contrast, effective robustness is barely affected on the natural splits (top row). This can be explained by the final data distributions of the different training splits: Our filtering procedure does not affect natural images which are most responsible for the performance on natural datasets which explains the consistency in performance.

We also investigate effective robustness on the DomainNet shifts in Fig. 18. We note that the ImageNet model's accuracy numbers on DomainNet are not comparable to the CLIP models because the ImageNet model has been evaluated on a subset of DomainNet (ImageNet-D, Rusak et al., 2022) which is compatible with ImageNet classes. DomainNet has many classes which are not present in ImageNet, such as for example "The Great Wall of China" or "paper clip" which have been removed in ImageNet-D to enable evaluating ImageNet trained models without the need for training an additional readout layer. In contrast, we evaluate the CLIP trained models on the full DomainNet splits following standard zero-shot evaluation procedure. We will add a Figure where we control for the missing classes and evaluate the CLIP models on ImageNet-D in the next version of the manuscript.

On DomainNet, we similarly observe strong changes in effective robustness of the CLIP trained models when evaluating on the stylistic domains (all domains except for DomainNet-Real), and barely any changes when evaluating on the DomainNet-Real domain.

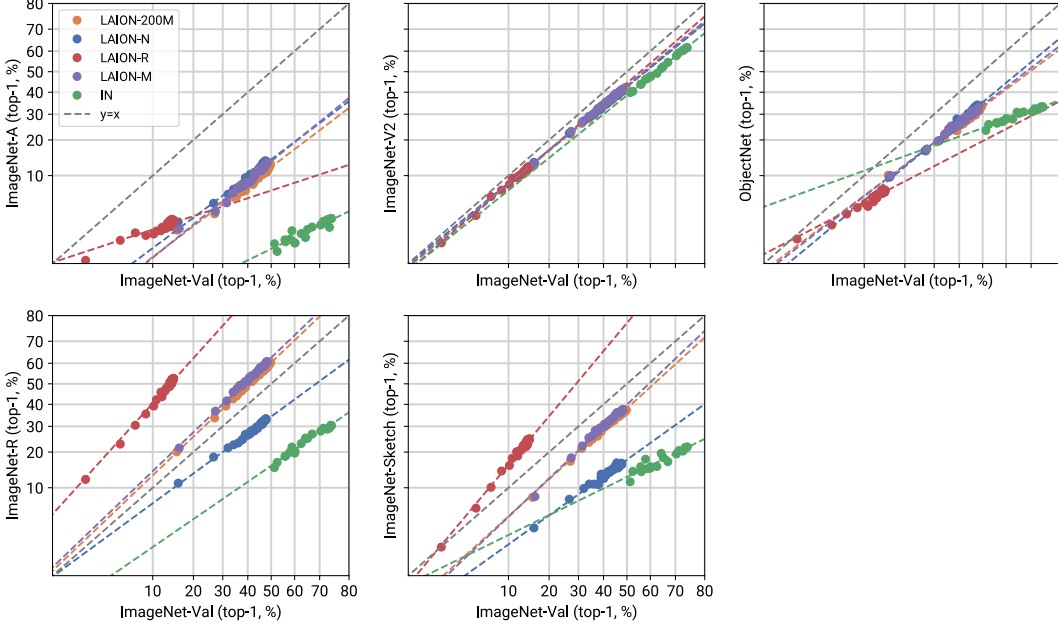

Figure 17: **Effective Robustness of different models on different ImageNet distribution shifts.** On ImageNet-R and ImageNet-Sketch (bottom row), we observe that the effective robustness of the CLIP models can be modulated by training it on the different dataset splits, i.e. LAION-Natural, LAION-Rendition, LAION-Mix. The model trained on LAION-Natural is much closer to the ImageNet trained model in terms of effective robustness compared to the LAION-Rendition model.

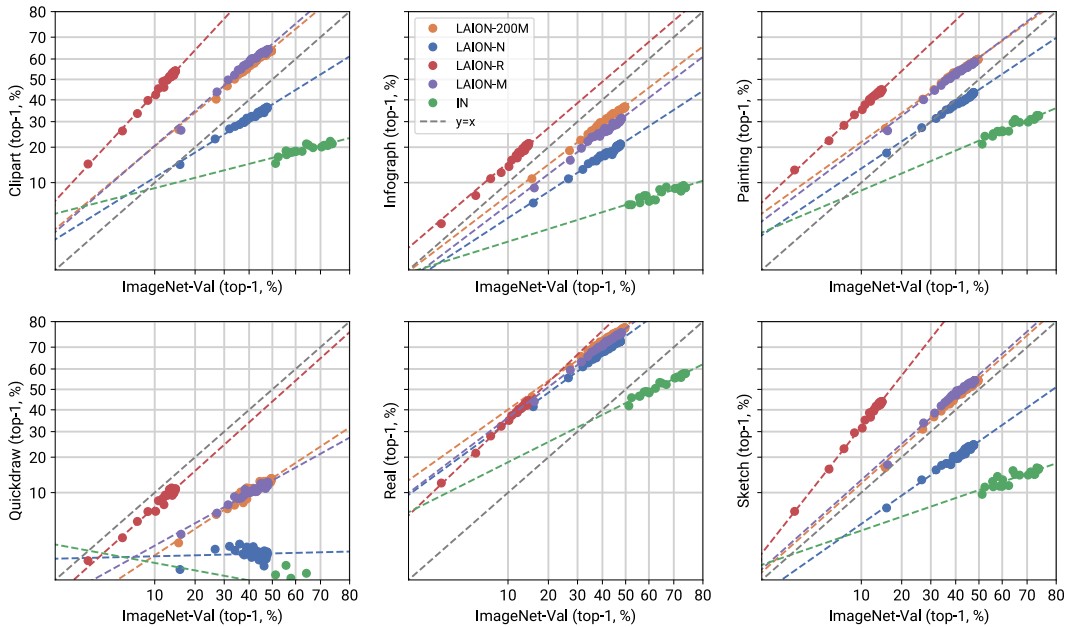

Figure 18: **Effective Robustness of different models on different DomainNet distribution shifts.**
On the stylistic domains, we observe that the effective robustness of the CLIP models can be modulated by training it on the different dataset splits, i.e. LAION-Natural, LAION-Rendition, LAION-Mix. Effective robustness barely changes when evaluating different CLIP models on DomainNet-Real.

## A.5 VISUALIZATION OF ERRORS MADE BY THE DOMAIN CLASSIFIER

We show images which have been misclassified by our domain classifier Fig. 19. We observe that the errors are interpretable. For example, the "natural" images which have been classified as "ambiguous" are indeed ambiguous: We see a sculpture in one image, a large woodwork of an ant in another and a pencil drawing of an airplane with a partly visible human hand drawing it in a third image.

## A.6 VISUALIZATION OF SAMPLES FROM THE LAION DATASET

We visualize random examples from the "Natural", "Rendition" and "Ambiguous" domains from LAION in Figs. 20 and 22.

## A.7 VISUALIZATIONS OF IMAGENET DISTRIBUTION SHIFTS

We visualize random examples from the "Natural", "Rendition" and "Ambiguous" domains from the considered ImageNet shifts datasets in Figs. 23 and 28. We show 20 images per split; occasionally, there are fewer than 20 images in some of these splits, such as e.g. there are very few renditions in ImageNet-A. In that case, we plot all images from that split and leave the remaining subplots blank.

## A.8 VISUALIZATIONS OF DOMAINNET DISTRIBUTION SHIFTS

We visualize random examples from the "Natural", "Rendition" and "Ambiguous" domains from different DomainNet datasets in Figs. 29 and 34. We show 20 images per split; occasionally, there are fewer than 20 images in some of these splits, such as e.g. no natural images in the Quickdraw domain. In that case, we plot all images from that split and leave the remaining subplots blank.

## A.9 EXTENDED DISCUSSION

**Object class distribution of our subsampled datasets** Our domain classifier separates images into three categories: *natural* images, *renditions*, and *ambiguous* images. While our classifier's accuracy and recall are high, it should be noted that we did not further control for potential biases (like favoring

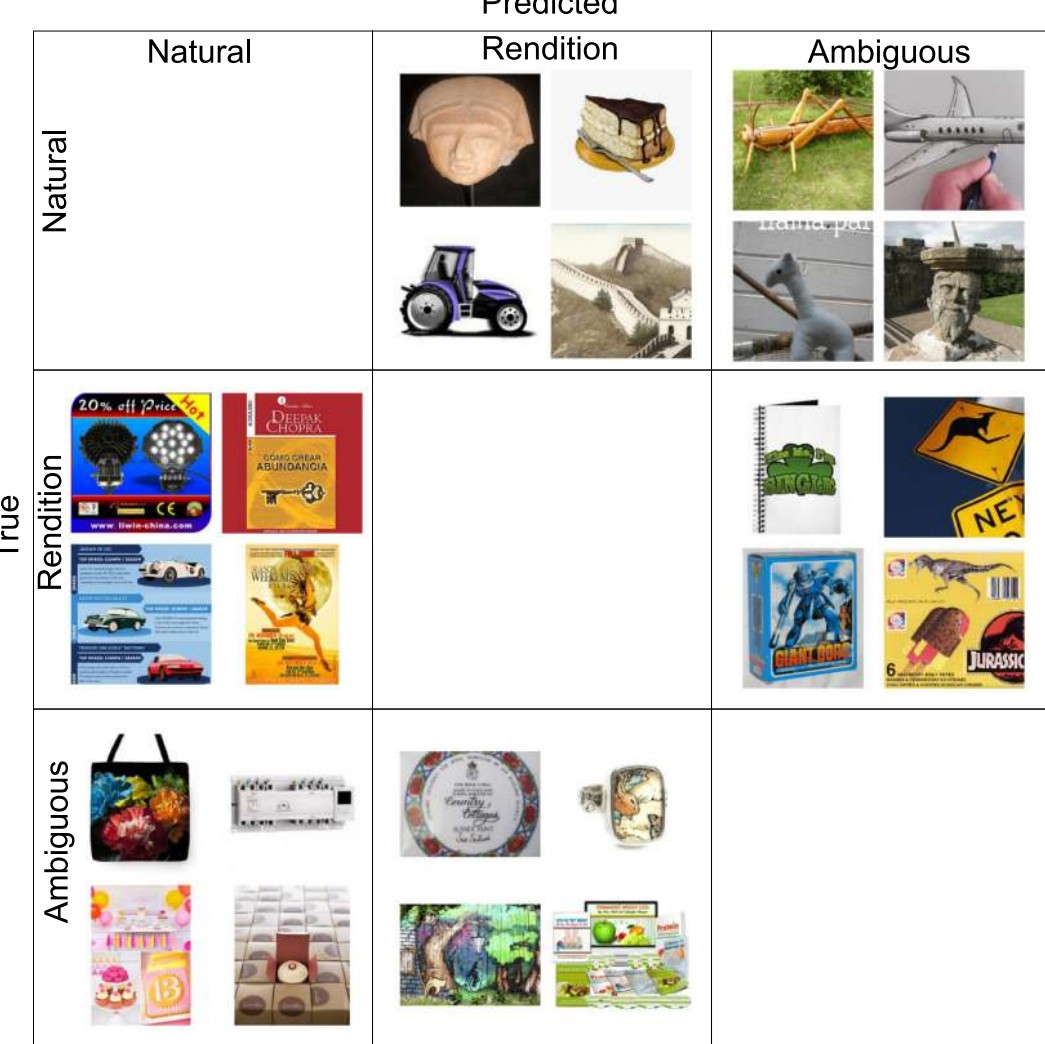

Figure 19: **Confusion matrix of example images which have been misclassified by our domain classifier.**

specific object classes within domains) or the overall object class distribution across all training and test sets. We therefore expect a dissimilar distribution of object classes in LAION-Natural and LAION-Rendition, and we leave a controlled analysis for future work.

**Ambiguous datapoints**   Our work does not examine the impact of ambiguous samples that exhibit both *natural* and *rendition* elements. To gain a clearer understanding of their effect, it is essential to distinguish between those ambiguous samples and those that belong to neither domain. We anticipate that the former category significantly enhances performance and sample efficiency, while the latter does not contribute substantially. A more thorough analysis of this distinction is left for future work.

**Short-cut learning**   The domain generalization gap in ImageNet models has been linked to shortcut learning, where models rely on features like texture over shape (Geirhos et al., 2018; 2019; 2020). While larger datasets are thought to mitigate this, our results suggest that simply adding more natural samples is insufficient to address all effects.

**Bias due to labeling**   Human labeling biases can propagate to classifiers and influence results. To address this, we rely on high-precision domain classifiers to filter millions of samples, minimizing domain contamination and ensuring the robustness of our conclusions. This approach balances scalability with accuracy while acknowledging the limitations of large-scale annotation.

**Efficacy of the domain classifiers**    The domain classifiers used in this work were trained, validated, and tested on randomly sampled subsets of LAION-200M, ensuring no distribution shift between their training and evaluation data. To ensure high reliability, the classifiers were deployed with a threshold of 98% precision, achieving strong precision and recall metrics on both the LAION-200M test set and test sets from ImageNet and DomainNet, as detailed in  Tabs. 1 and 5. Additionally, random samples from the classified LAION-Natural and LAION-Rendition datasets, visualized in Figs. 3, 21 and 22, confirm that the retrieved samples align well with their respective natural or rendition categories. Finally, our core results demonstrate that models trained on these subsets excel in their respective domains but show limited performance on the other, further validating the effectiveness of the classifiers in accurately separating the domains.

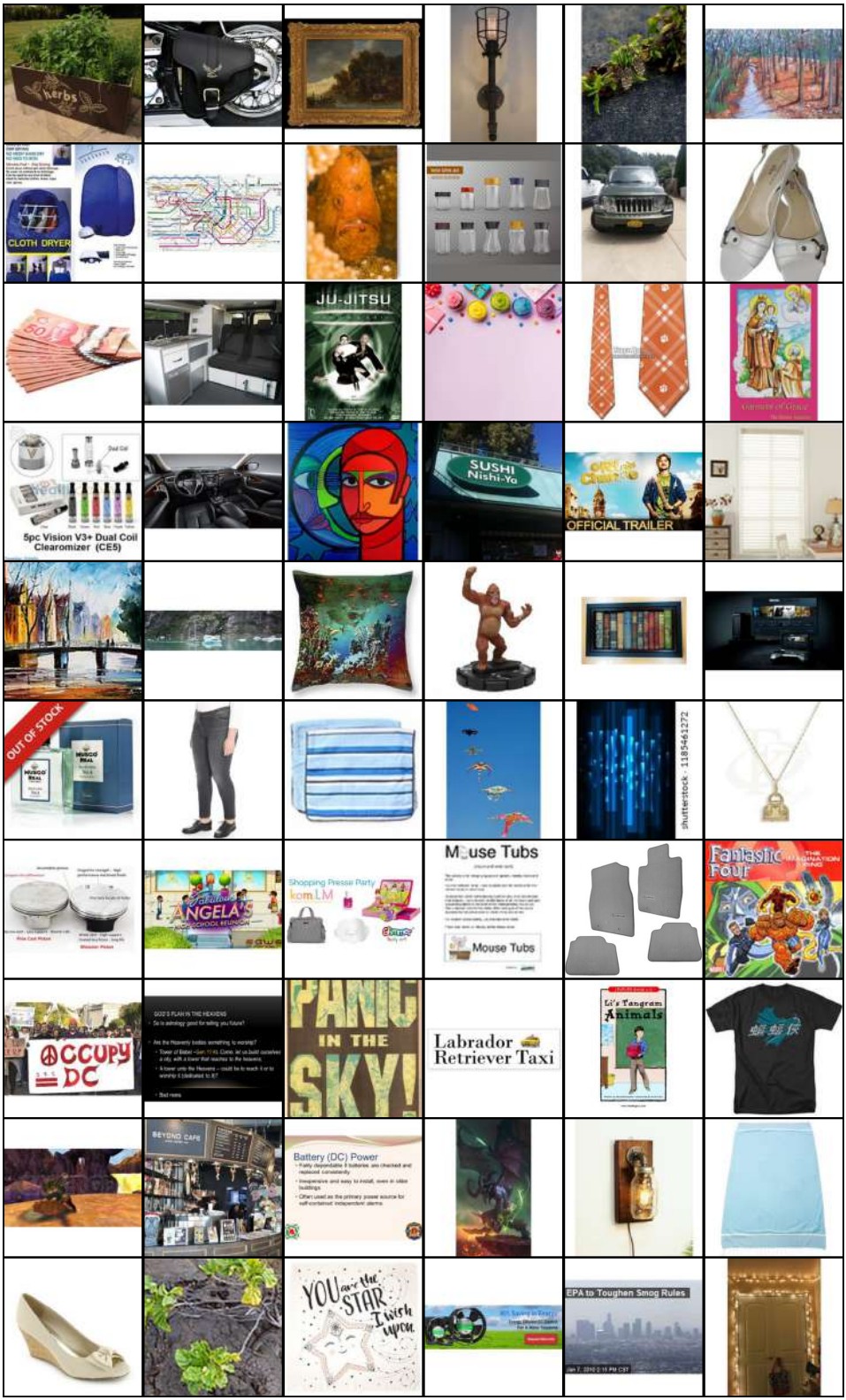

Figure 20: **Random samples from LAION-200M**. We omit NSFW images and images of humans.

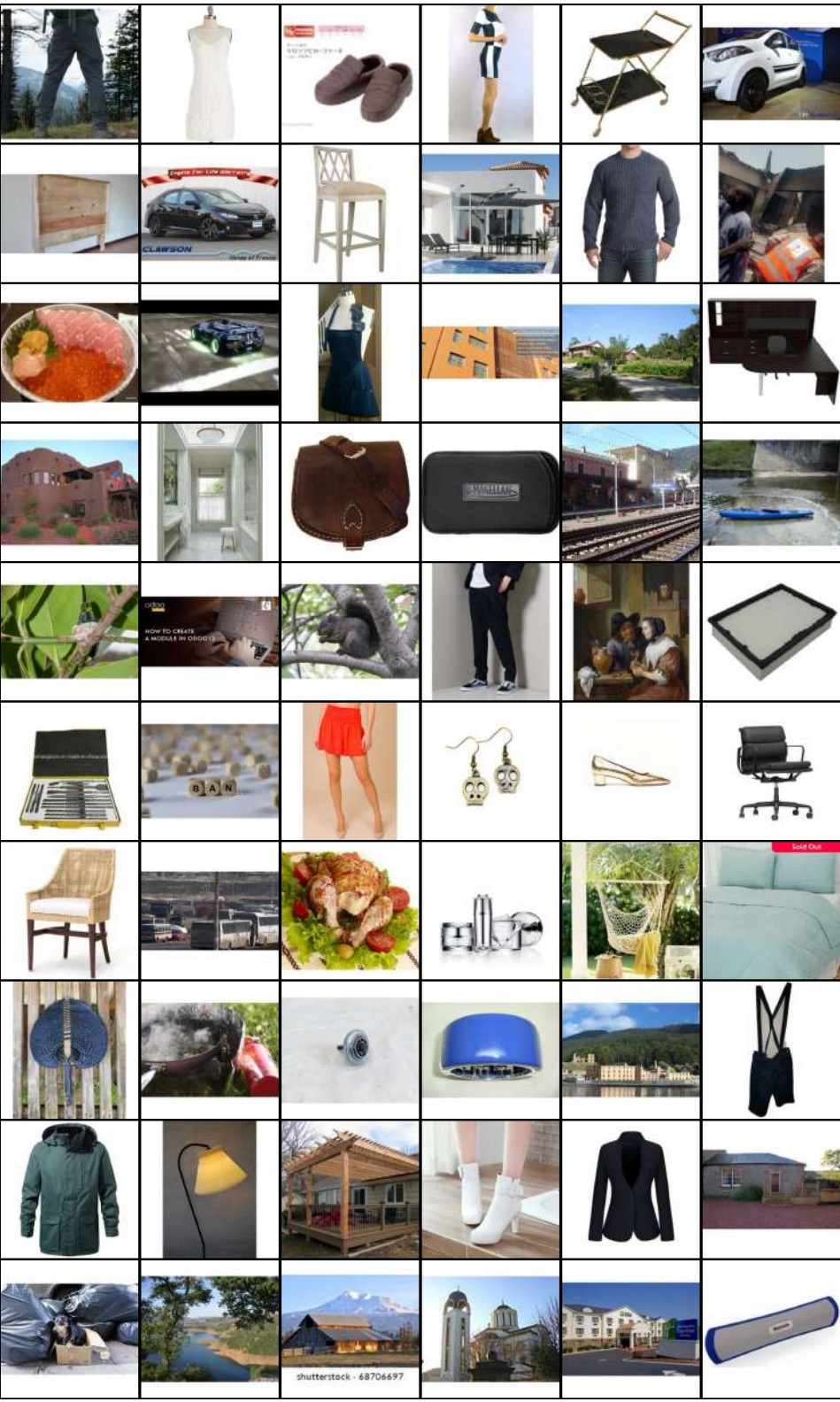

Figure 21: **Random samples from LAION-Natural**. We omit NSFW images and images of humans.

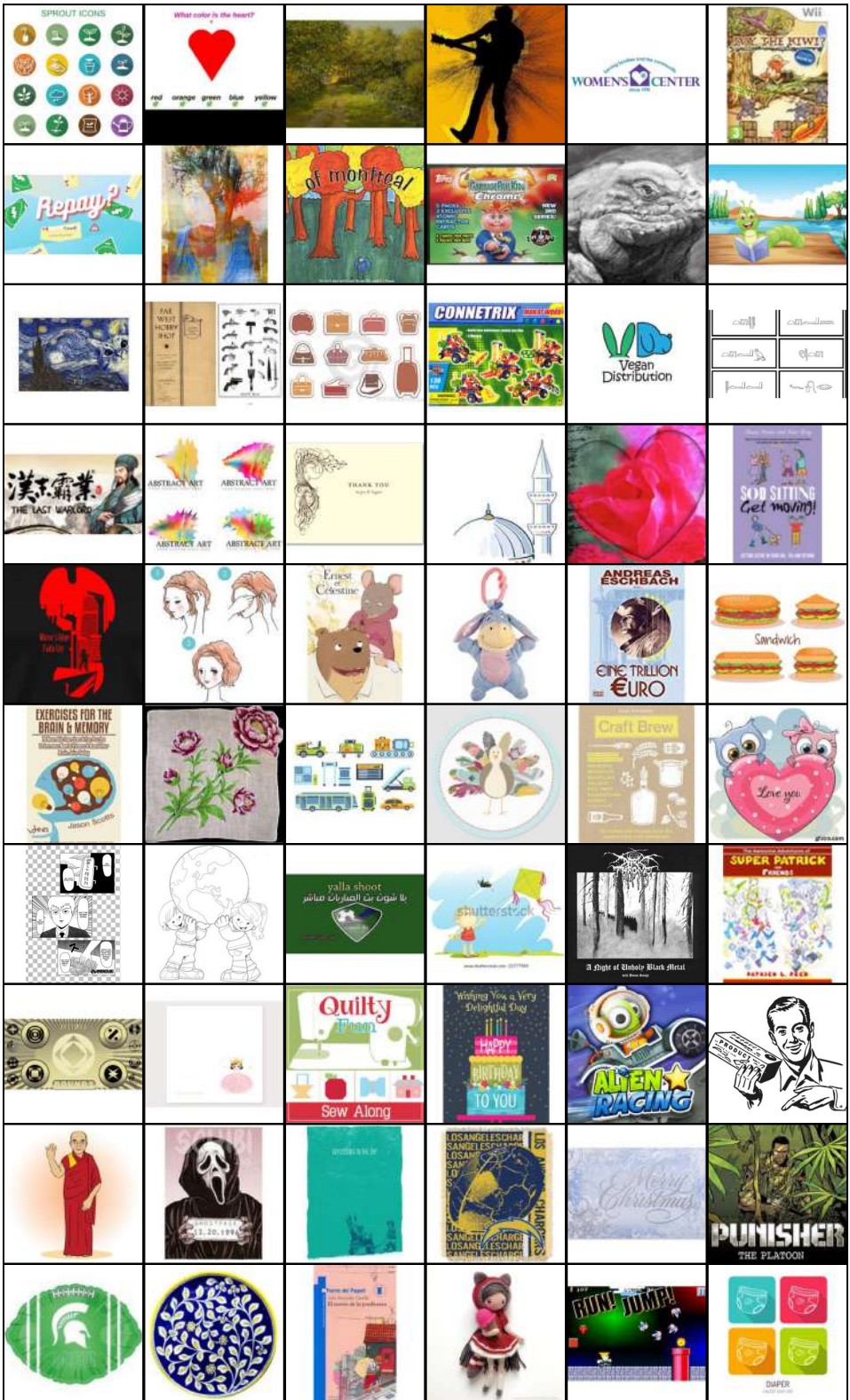

Figure 22: **Random samples from LAION-Rendition**. We omit NSFW images and images of humans.

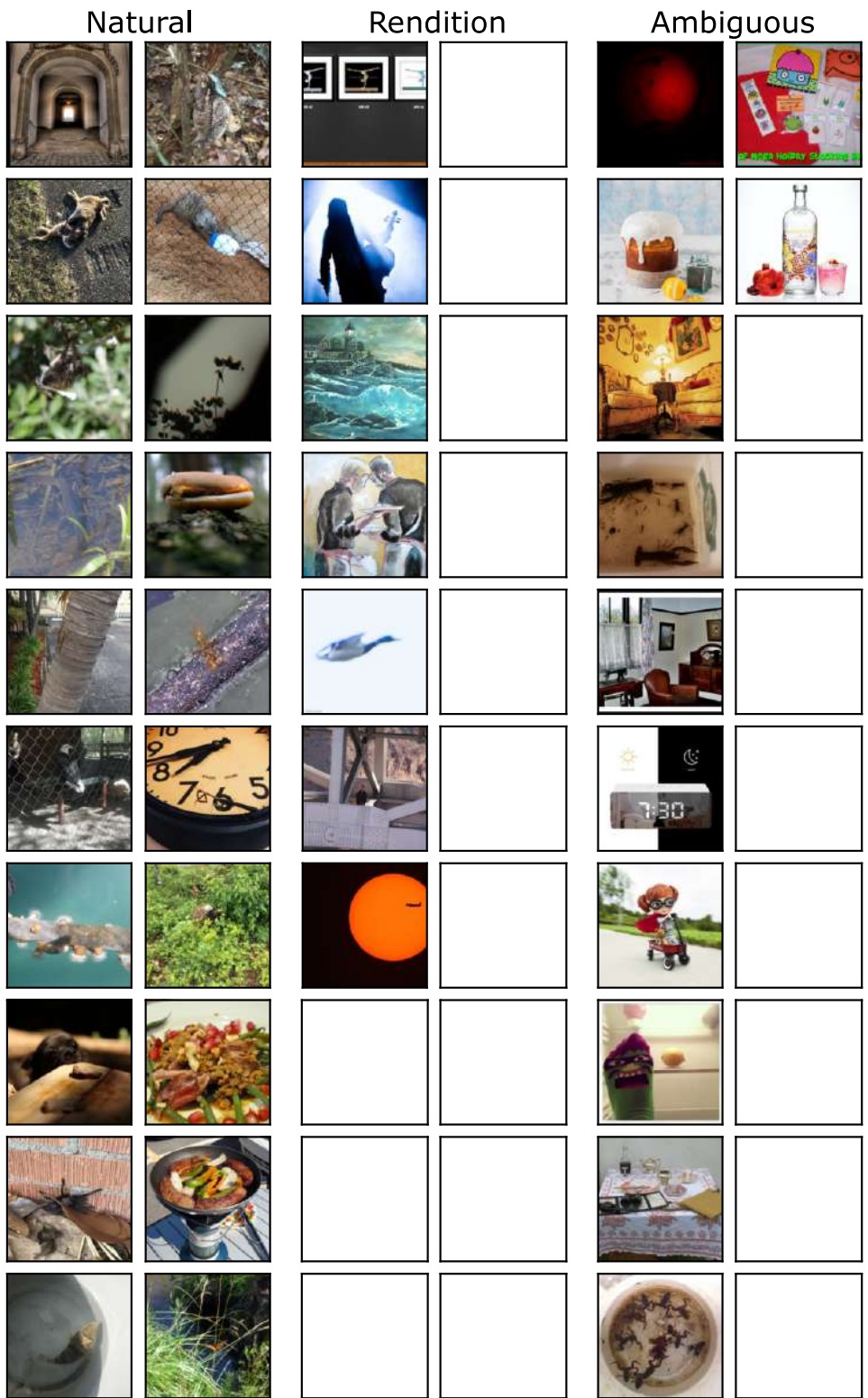

Figure 23: **Random samples of ImageNet-A grouped by domain.** We omit NSFW images and images of humans.

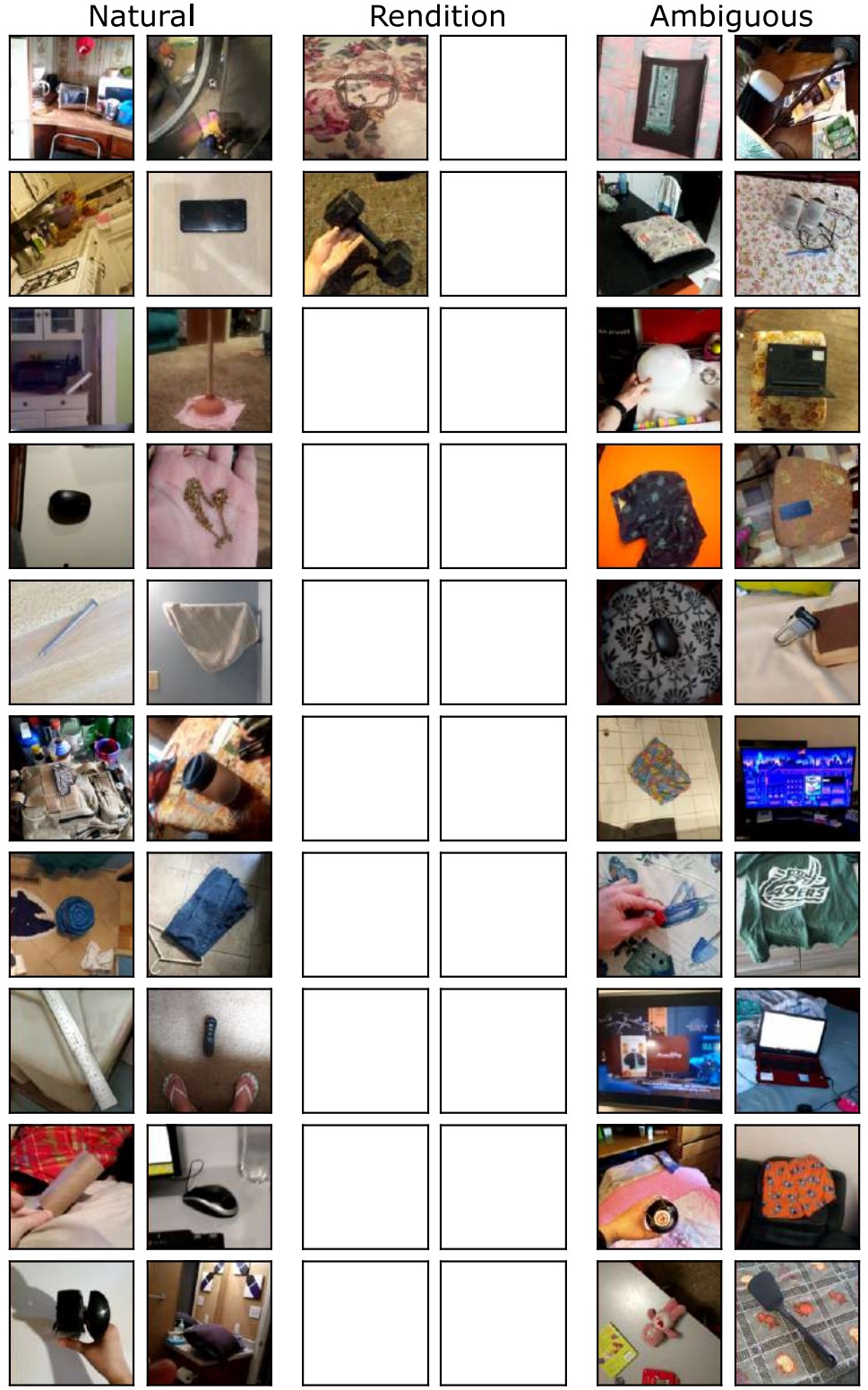

Figure 24: Random samples of ObjectNet grouped by domain. We omit NSFW images and images of humans.

Natural          Rendition          Ambiguous

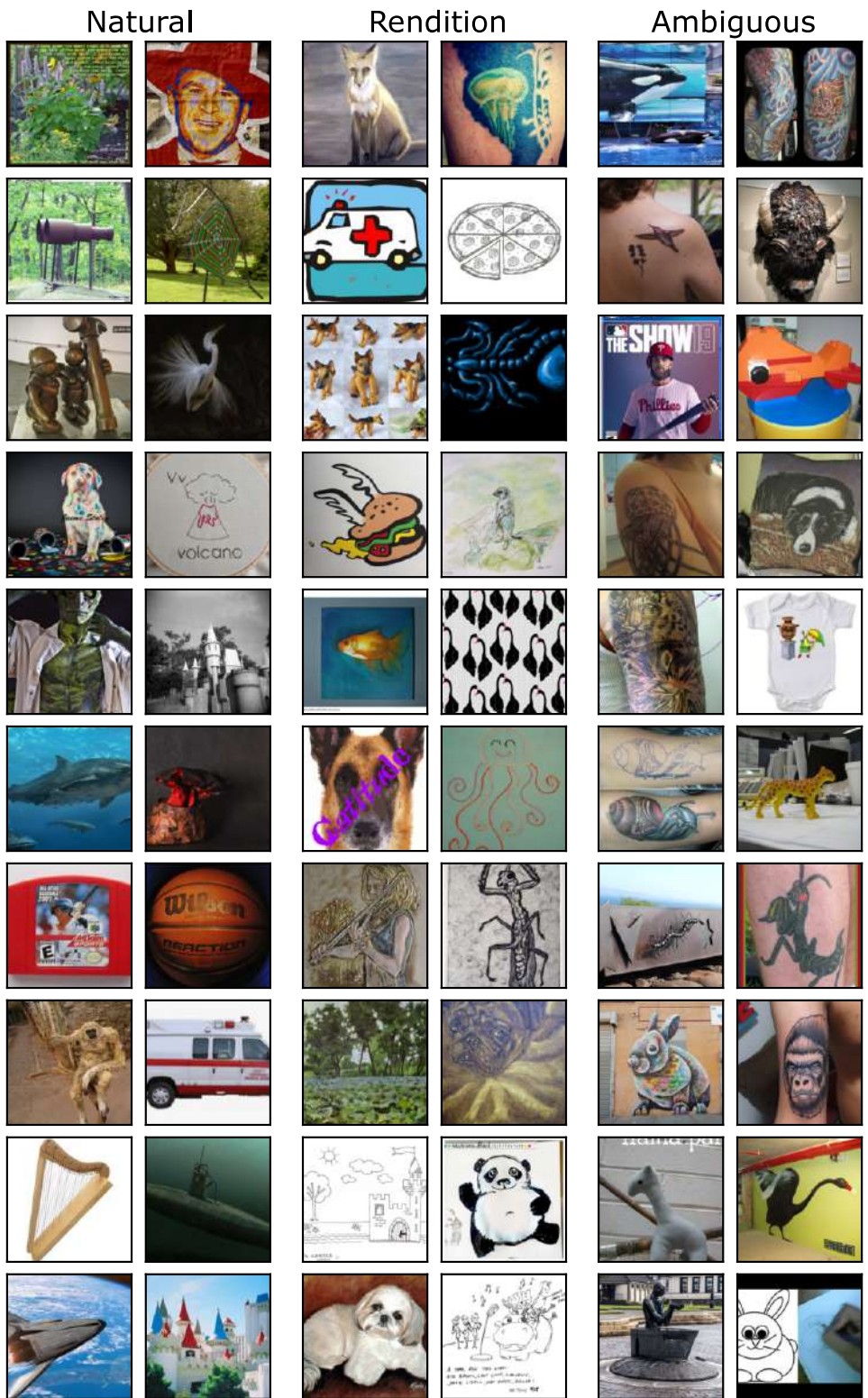

Figure 25: **Random samples of ImageNet-R grouped by domain.** We omit NSFW images and images of humans.

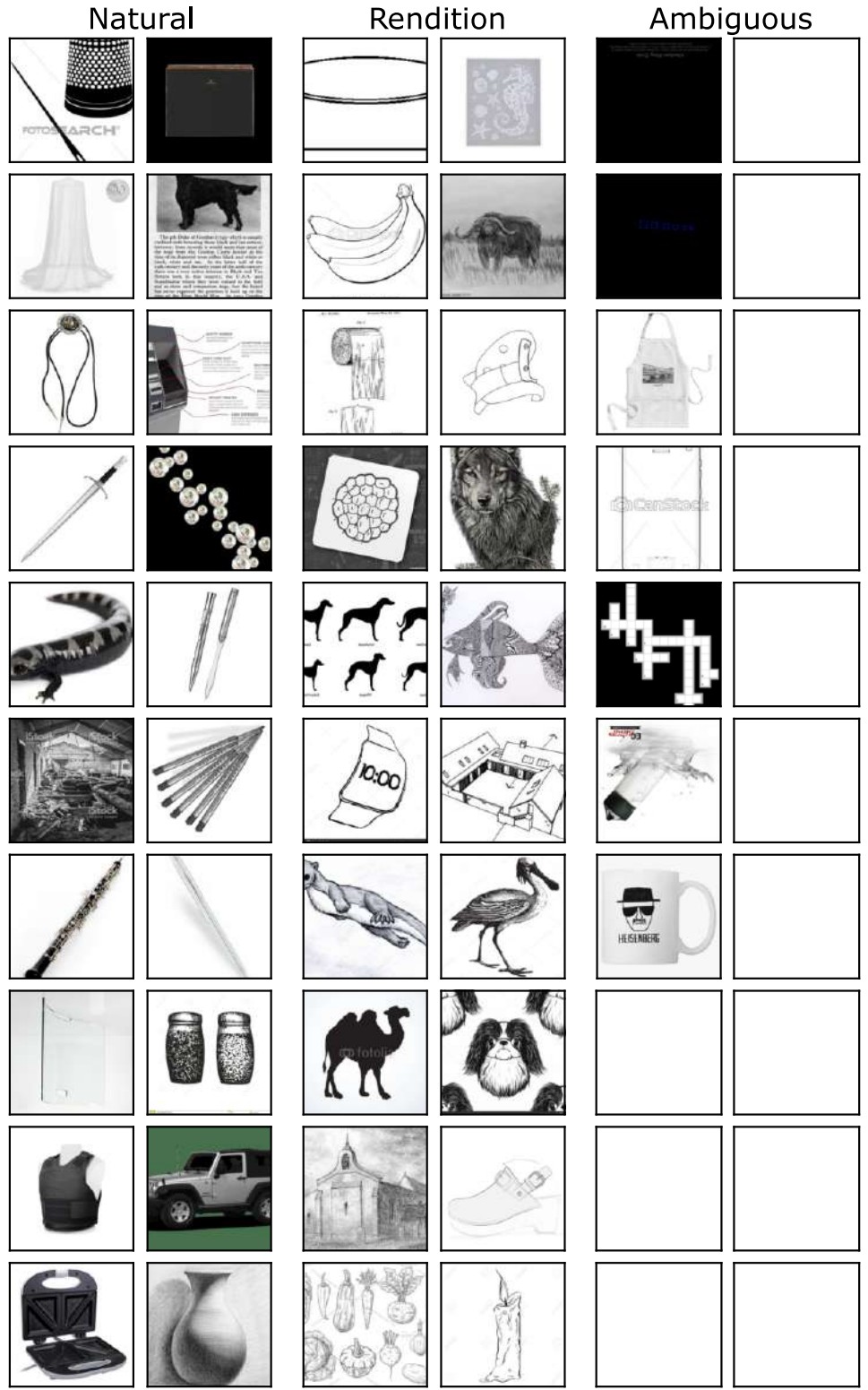

Figure 26: **Random samples of ImageNet-Sketch grouped by domain.** We omit NSFW images and images of humans.

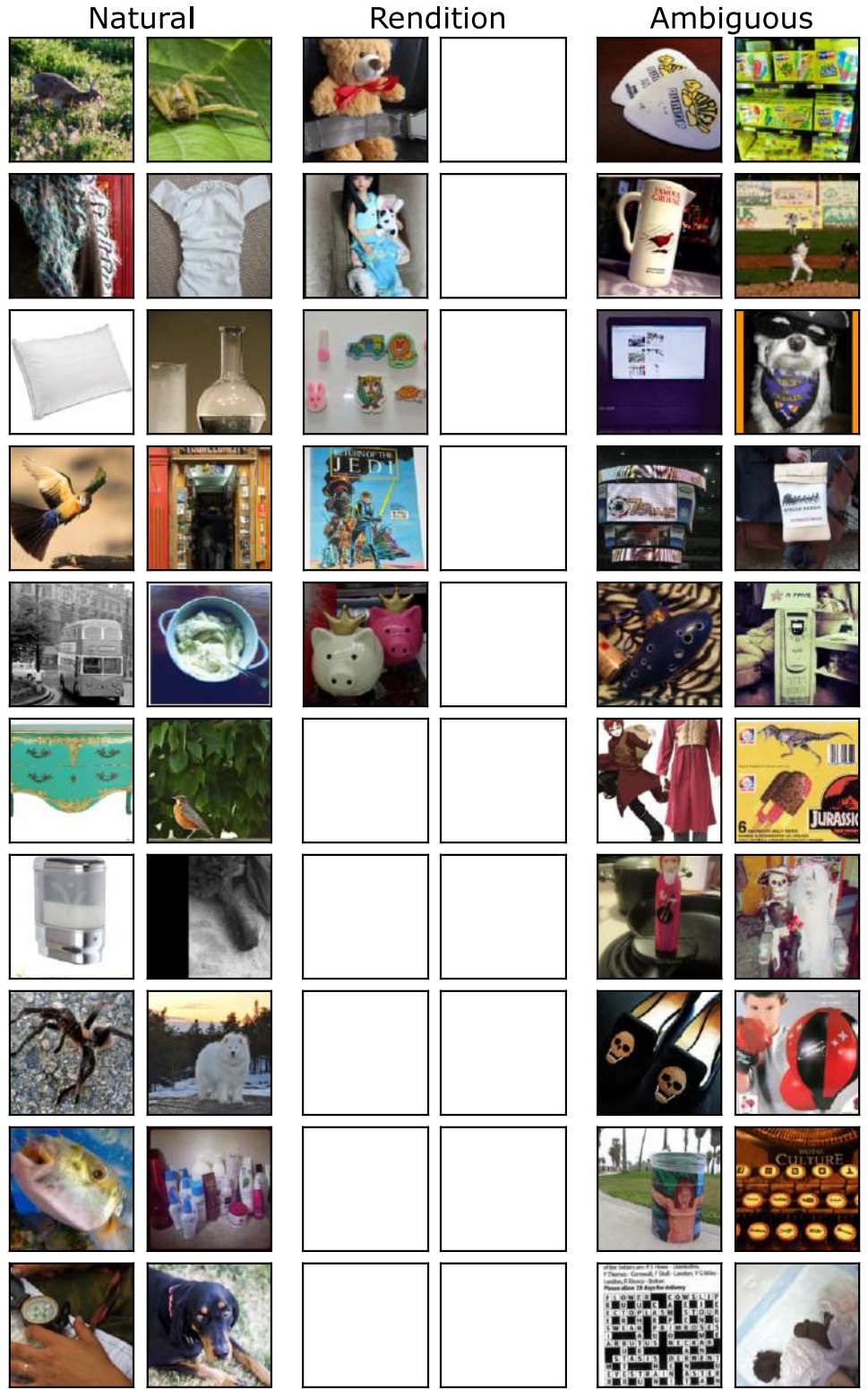

Figure 27: **Random samples of ImageNet-V2 grouped by domain.** We omit NSFW images and images of humans.

Natural                Rendition                Ambiguous

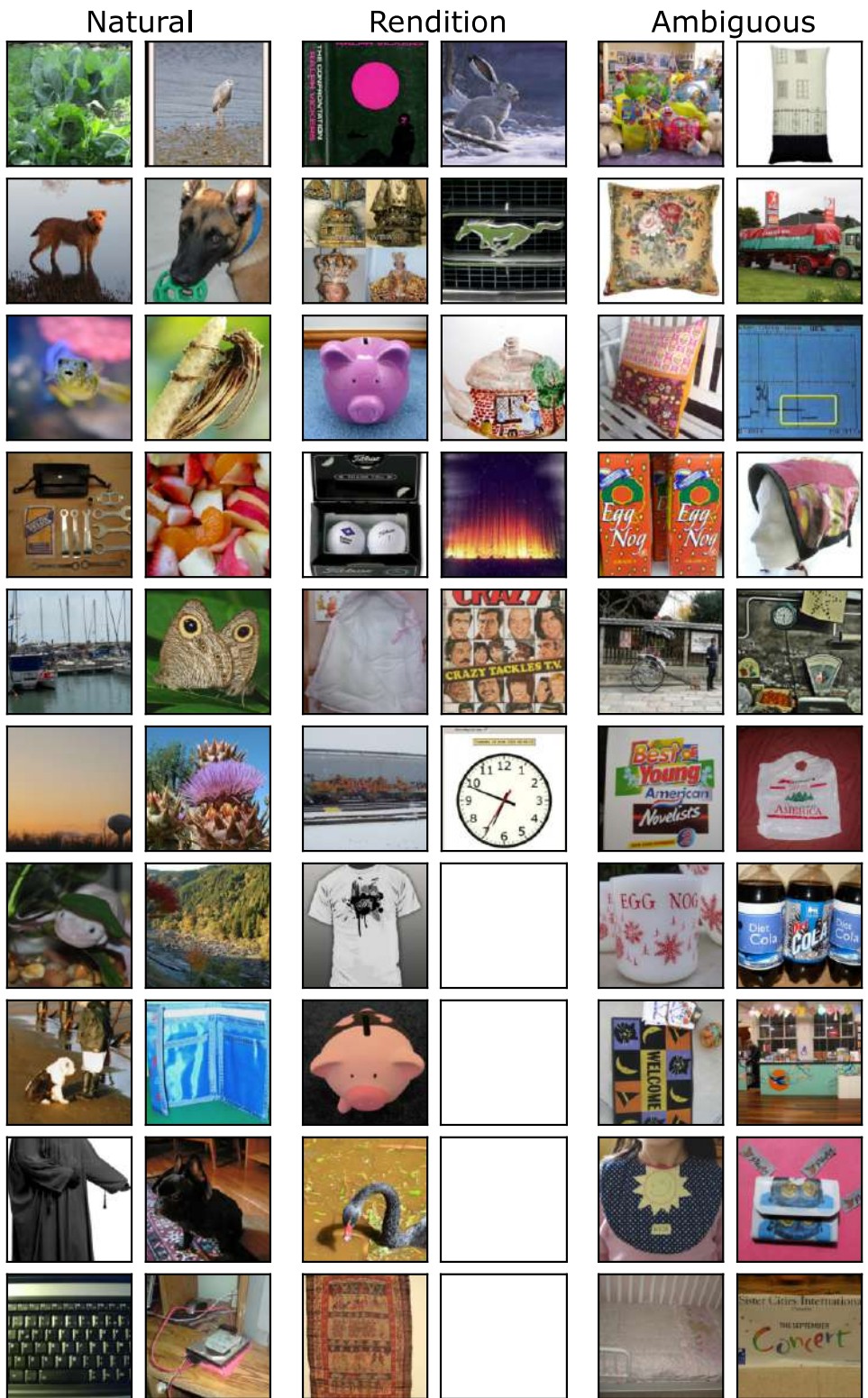

Figure 28: **Random samples of ImageNet-Val grouped by domain.** We omit NSFW images and images of humans.

Natural          Rendition          Ambiguous

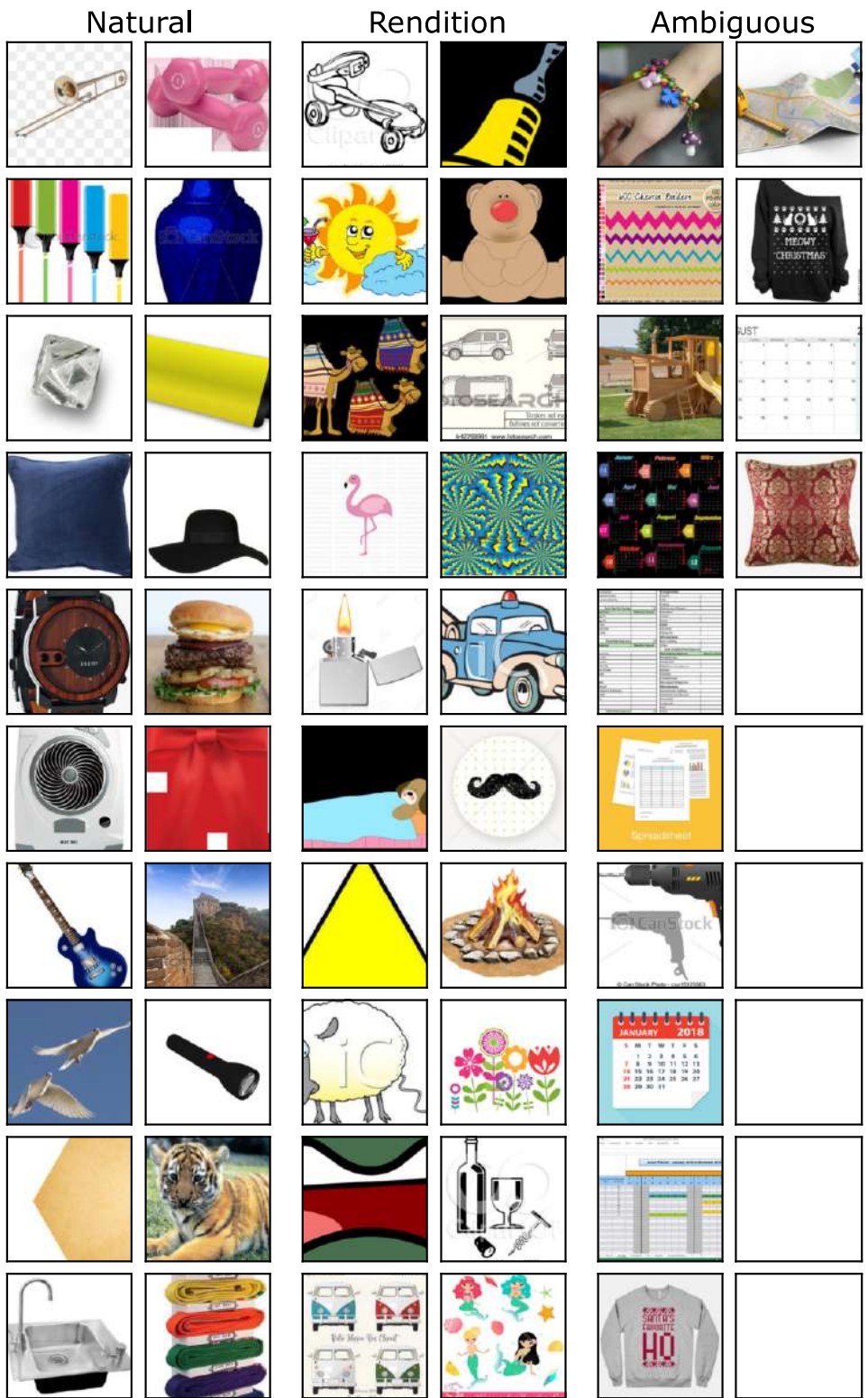

Figure 29: **Random samples of DomainNet-Clipart grouped by domain.** We omit NSFW images and images of humans.

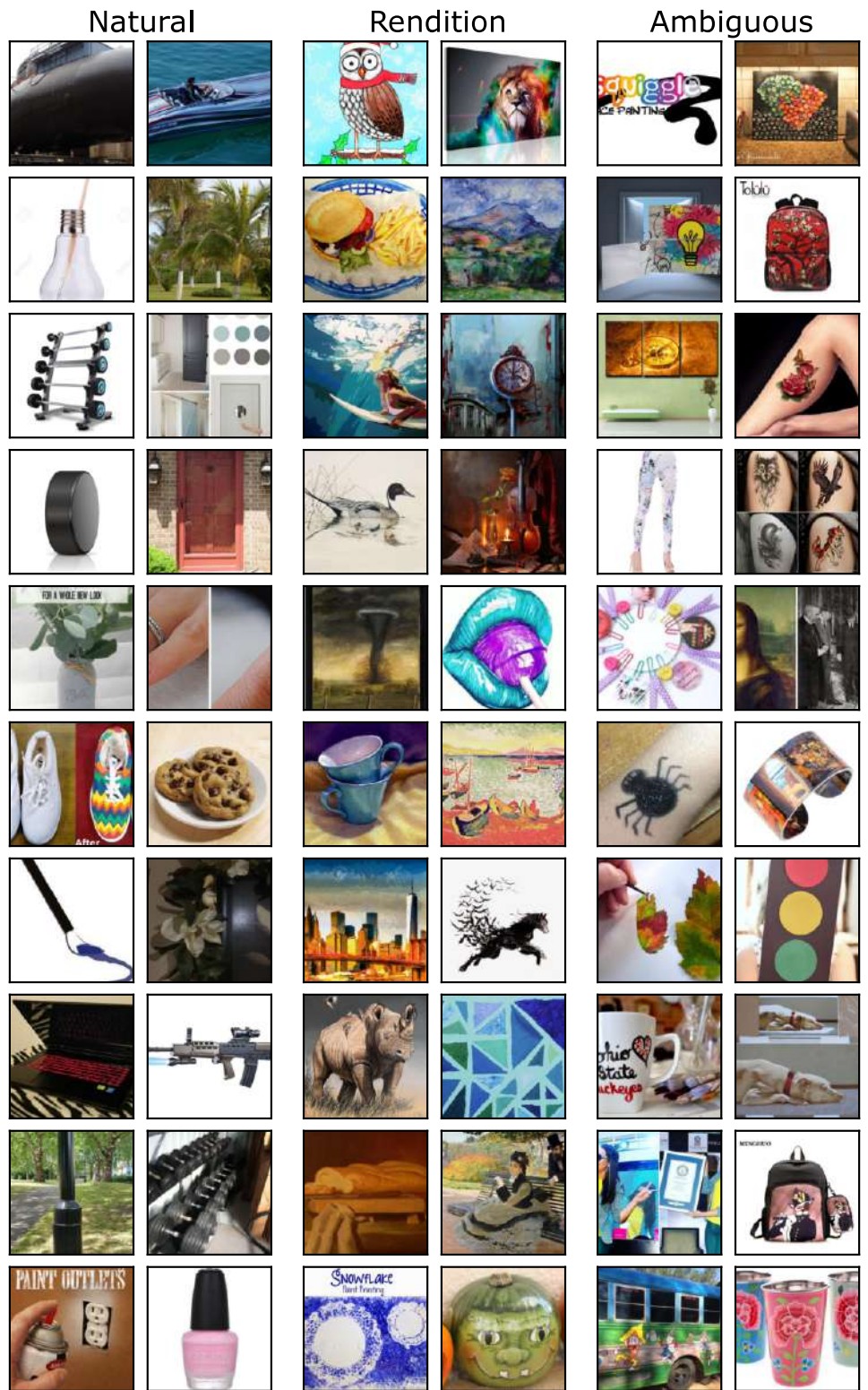

Figure 30: **Random samples of DomainNet-Painting grouped by domain.** We omit NSFW images and images of humans.

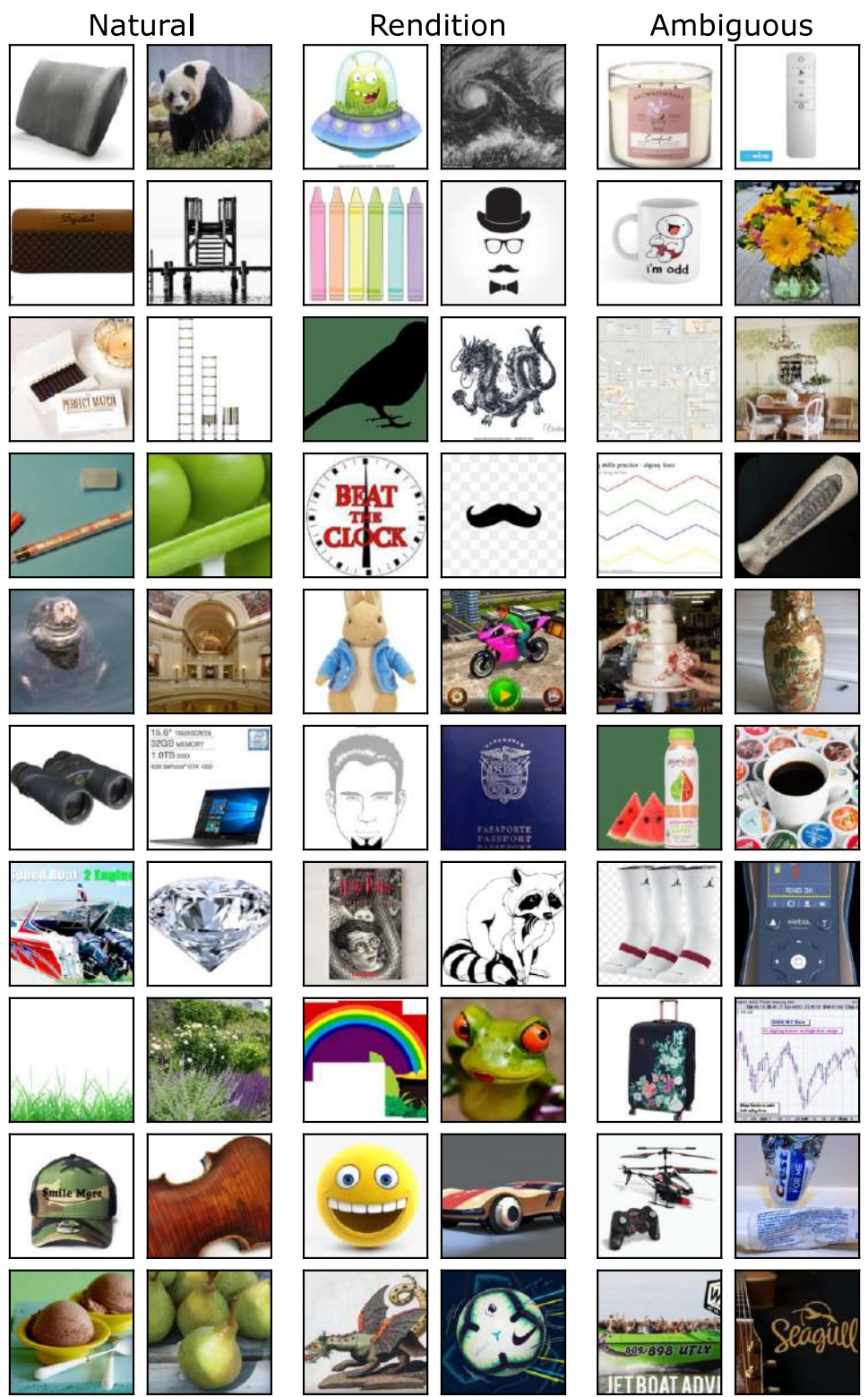

Figure 31: **Random samples of DomainNet-Real grouped by domain.** We omit NSFW images and images of humans.

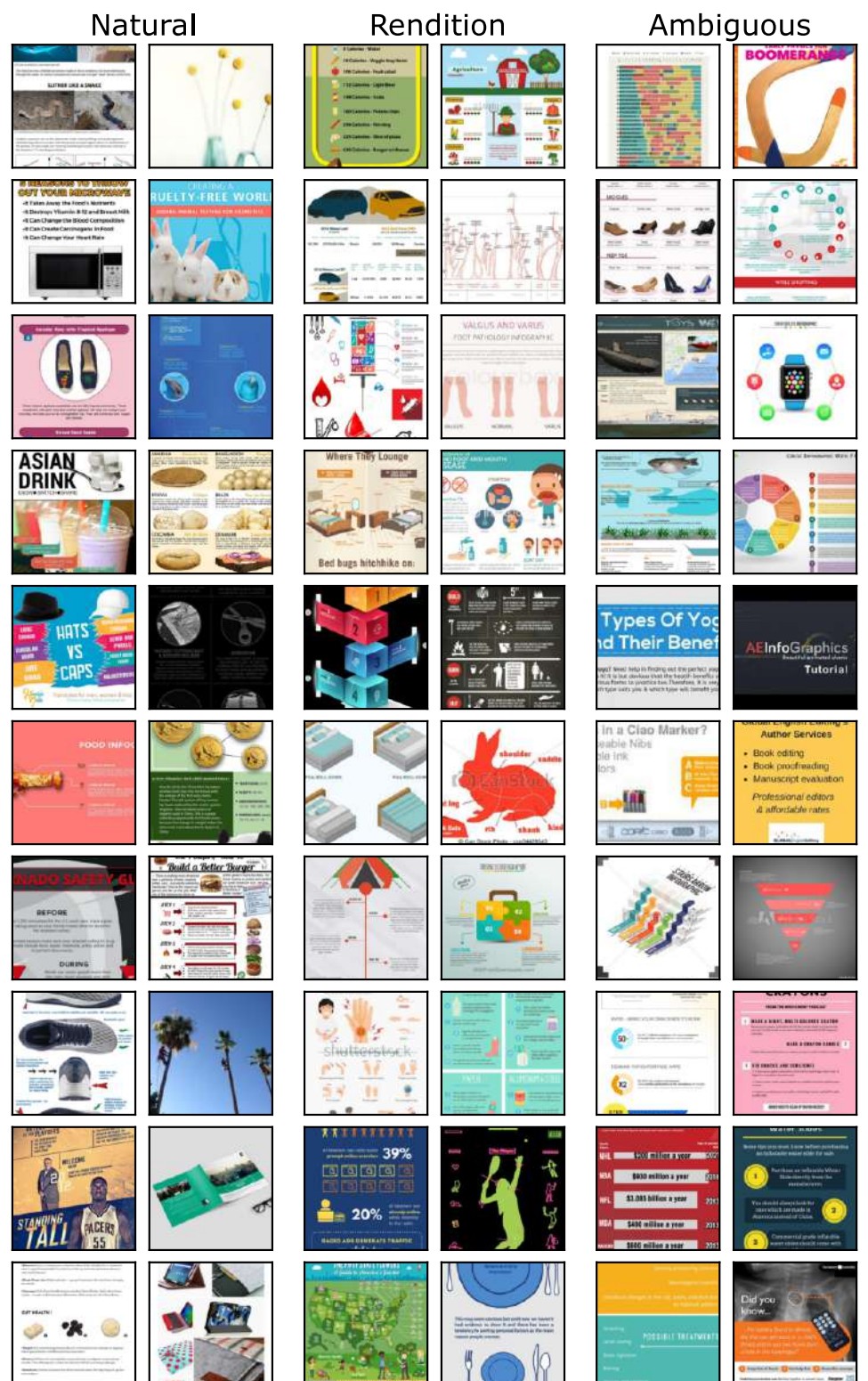

Figure 32: **Random samples of DomainNet-Infograph grouped by domain.** We omit NSFW images and images of humans.

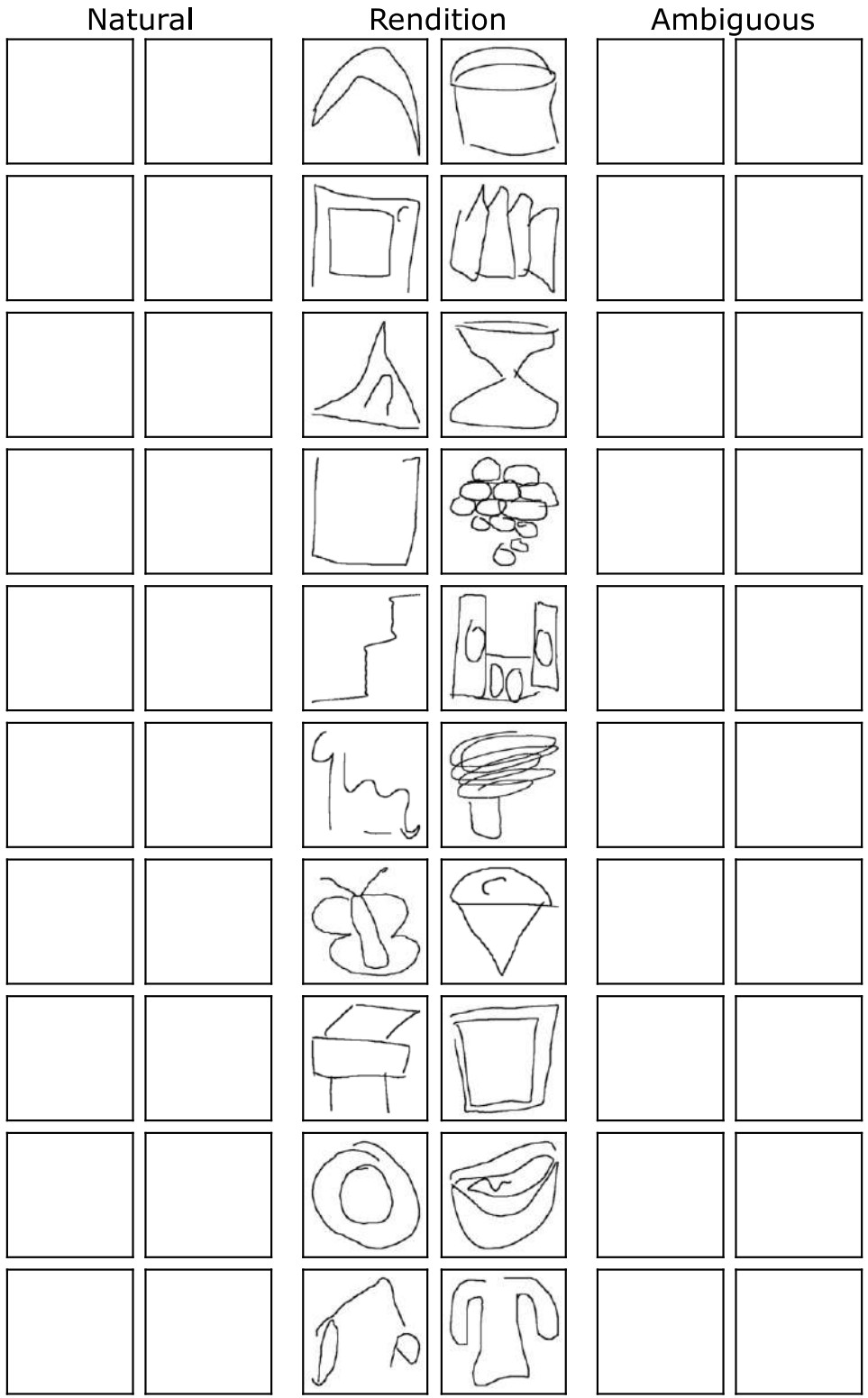

Figure 33: **Random samples of DomainNet-Quickdraw grouped by domain.** We omit NSFW images and images of humans.

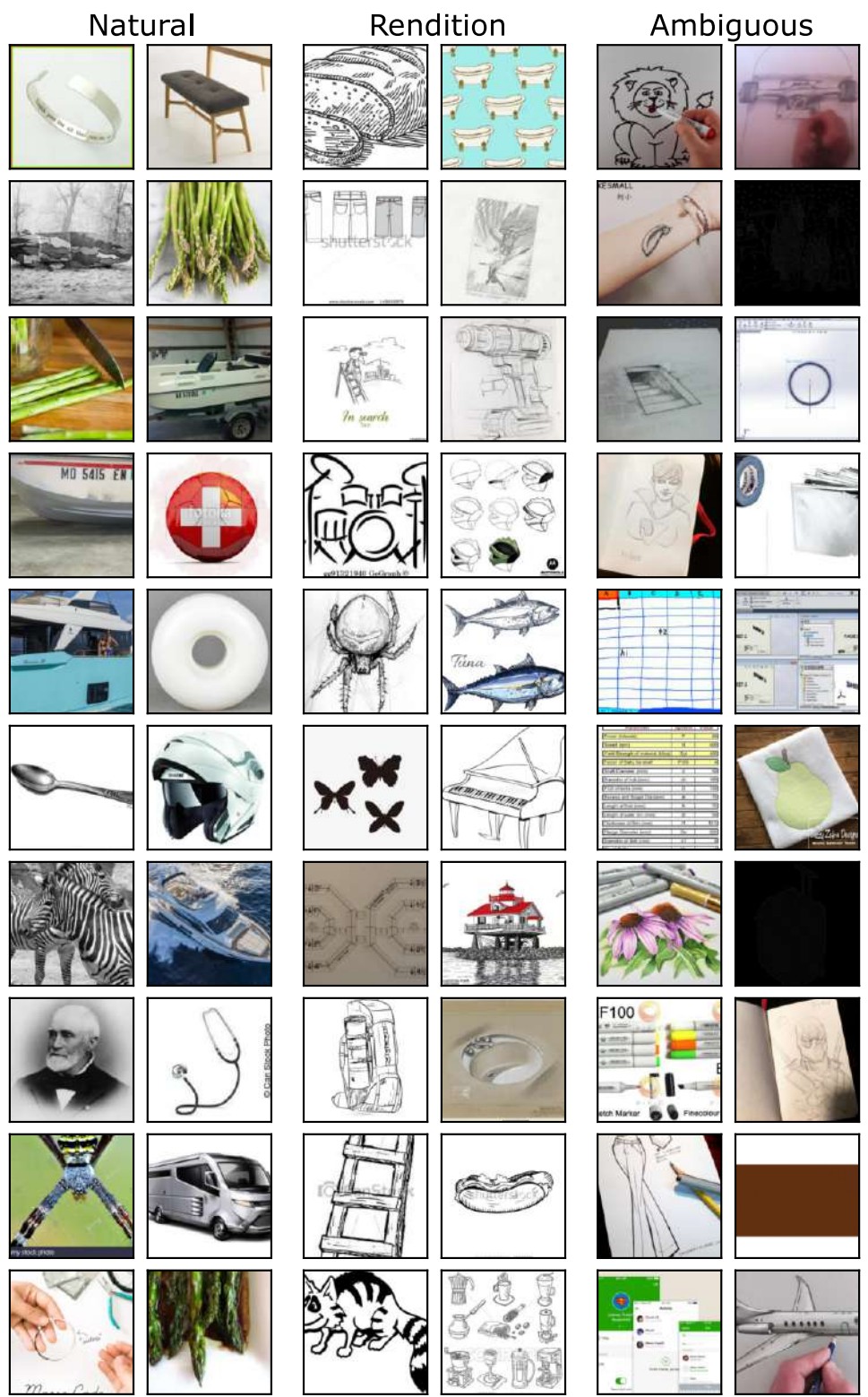

Figure 34: **Random samples of DomainNet-Sketch grouped by domain.** We omit NSFW images and images of humans.

