# OpenReview forum: "In Search of Forgotten Domain Generalization"
_ICLR.cc/2025/Conference — ICLR 2025 Spotlight_

### Official Review · Reviewer_8EXX · 2024-10-18

**Soundness:** 3
**Presentation:** 3
**Contribution:** 2
**Rating:** 8
**Confidence:** 4

**Summary:**

The paper focuses on exploring the phenomenon of test domain contamination in the era of foundation models. The paper constructs two different style datasets, LAION-Natural and LAION-Rendition, which are strictly OOD relative to the corresponding ImageNet and DomainNet test sets in terms of style. It examines whether the good OOD performance of CLIP results from test set contamination or from its training paradigm. The findings indicate that CLIP's success stems from domain contamination rather than an intrinsic OOD generalization ability. Additionally, the authors suggest how the mixing ratio of Rendition and Natural samples in the training set can optimize model performance on the test set.

**Strengths:**

1. This paper gives a very detailed analysis that clarifies that the success of CLIP mainly comes from the training data rather than its own training paradigm, which can give some inspiration to many researchers.

2. The authors give the optimal mixing ratio of rendition and natural image, which allows for better generalization performance of models trained on this dataset.

3. The authors conduct extensive experiments and analyses to support the claim that "the success of the base model in OOD settings does not originate from its own characteristics", which enhances its credibility.

**Weaknesses:**

1. The main issue with this paper is that it appears to draw a conclusion "he success of CLIP mainly comes from the training data rather than its own training paradigm" that lacks sufficient insight for subsequent researchers. While the question of the optimal ratio of natural and rendition images in the training set is indeed illuminating, the authors do not seem to have conducted tests with models of different architectures. Additionally, I believe that this experiment is important because it can ensure that the optimal ratio is not sensitive to the model's architecture.

2. Although I have experience in OOD research, I am not up to date with the latest techniques and conclusions in this area. Therefore, I will be observing the perspectives of other reviewers as well as the authors' responses. Overall, the paper was thorough and solid, but it seemed to lack novelty.for me.

**Questions:**

No

---

> ### Author Response · Authors · 2024-11-18
>
> Dear Reviewer 8EXX,\
> Thank you for acknowledging our paper’s strength, helpful comments, and feedback. We address each of your concerns below:
>
> **[Clarification of core insight]:** Please note that before, numerous studies implicitly or explicitly assume that CLIP generalizes OOD due to inherent robustness or emergent properties of large training datasets. This is reflected in the following quotes:
> - “zero-shot transfer is more an evaluation of CLIP’s robustness to distribution shift and domain generalization” [1]
> - “several studies suggested that some Vision-Language Models (VLMs) such as the CLIPs, exhibit OoD generalization” [2]
> - “VLP (Vision-Language Pretraining) models not only show good performance in-distribution (ID) but also generalize to out-of-distribution (OOD) data” [3]
> - “models such as CLIP, ALIGN, and BASIC have recently demonstrated unprecedented robustness on a variety of natural distribution shifts” [4]
> - “These models (CLIP, GLIP, etc.) have demonstrated enormous potential in, a wide range of downstream applications… especially for the out-of-distribution (OOD) samples” [5]
> - “CLIP achieves OOD generalization in a task-agnostic way” [6].
>
> This means that our paper challenges a common belief through rigorous experiments: When truly OOD test data is used—by removing test domain samples from training—CLIP fails to maintain robustness, revealing a critical flaw in standard evaluations. This insight highlights the need for rigorous OOD evaluations and tempers expectations about CLIP’s true generalization capabilities. Given this context, we hope you can see the novelty of our work. To better emphasize this core insight, we have revised the main text (Lines 80–88) and included a more detailed discussion (Sec. 6 first paragraph).
>
> **[Usefulness for subsequent researchers]:** Our work not only provides key insights into dataset design and serves as a cautionary tale for OOD evaluations but also offers invaluable resources to the research community—a critical step toward improving model robustness. Specifically, we contribute:
> - the LAION-Natural and LAION-Rendition datasets, each containing tens of millions of images that are rigorously OOD with respect to ImageNet and DomainNet test sets in terms of style
> - domain classifier checkpoints
> - domain classifier training data
> - trained CLIP model checkpoints
>
> To our knowledge, this is the first effort to deliver pretraining datasets of such scale and domain specificity, addressing a major gap in OOD generalization research. The value of these contributions is reflected in reviewer feedback, with reviewer HGzr describing our work as tackling an *“extremely important problem”* with *“impactful findings,”* *“useful datasets,”* and *“useful methodology,”* while reviewer Gz5f highlights that the paper *“studies a valuable problem”* with *“sound analysis”* to support its claims.
>
> **[Sensitivity to model architecture]:** Prior work provides strong evidence that our data contamination findings and the optimal ratio findings transfer to other architectures and training methods, not just CLIP [4,7]. For instance, [4] *“systematically ruled out the training set size, language supervision, and the contrastive loss function as explanations for the large robustness gains achieved by the CLIP models [...] CLIP’s robustness is dominated by the choice of the training distribution, with other factors playing a small or non-existent role.”* They further show that models trained on identical data distributions but with varying loss functions (SimCLR+FT, CLIP, Supervised) or model architectures (different backbones, parameter sizes) all exhibit the same effective robustness. This suggests that our conclusions will hold broadly across model types. We have included a new paragraph in the discussion section to address this point (Sec. 6).
>
> We hope that our detailed explanations have adequately addressed your questions and concerns raised. We sincerely hope this information offers a clearer understanding of our work, allowing you to increase your score.
>
> [1] Radford et al, “Learning Transferable Visual Models From Natural Language Supervision” \
> [2] Abbasi et al, “Deciphering the Role of Representation Disentanglement: Investigating Compositional Generalization in CLIP Models” \
> [3] Nguyen et al, “SAFT: Towards Out-of-Distribution Generalization in Fine-Tuning” \
> [4] Fang et al, “Data Determines Distributional Robustness in Contrastive Language Image Pre-training (CLIP)” \
> [5] Li et al. “Distilling Large Vision-Language Model with Out-of-Distribution Generalizability” (2023) \
> [6] Shu et al, “CLIPood: Generalizing CLIP to Out-of-Distributions” \
> [7] Miller et al. Accuracy on the Line: On the Strong Correlation Between Out-of-Distribution and In-Distribution Generalization

---

> > ### Comment · Reviewer_8EXX · 2024-11-22
> >
> > Thank you to the author for the detailed explanation and response. I found it very enlightening, as I had overlooked the fact that this paper challenges a common belief through rigorous experimentation. I have decided to raise my score to 8. Thanks for the excellent work!

---

> > > ### Author Response · Authors · 2024-12-03
> > >
> > > Dear Reviewer 8EXX,
> > >
> > > Thank you for thoughtfully engaging with our work, posing clarifying questions, and considering our responses. We greatly appreciate your support and advocacy for our paper.

---

### Official Review · Reviewer_Gz5f · 2024-11-04

**Soundness:** 3
**Presentation:** 3
**Contribution:** 3
**Rating:** 8
**Confidence:** 4

**Summary:**

This paper reintroduces the OOD problem in the era of the foundation model. They propose the argument that the phenomenon that foundational models are no longer  susceptible to OOD samples is merely an illusion due to the *domain contamination*, which they define as "whether crucial aspects of a test domain are included in the training domain, e.g., by including images with different content but similar style to test samples"

They then introduce cleaned single-domain datasets to study this problem, discovering that CLIP trained only on natural images significantly underperforms on rendition domain shifts. They further investigate the domain mixing and scaling that affects OOD performance.

**Strengths:**

1. This paper studies a valuable problem that has seldom been studied. Indeed, the OOD problem, which seems to be less of a problem in the era of LLM, might be an illusion due to the scaling of the massive data. This problem should be taken seriously, as there are still areas where collecting data is extremely difficult and scaling will likely fail.
2. Sound analysis that supports the claim.

**Weaknesses:**

1. Figure 6 A is a bit confusing, as the result of the best rendition-to-natural ratio 1:3 and 1:1 can not be read in this figure. I suggest adding ratio labels to the color scale or individual data points in the figure.
2. Can you provide some discussion on the "true effectiveness" of the domain classifier trained and evaluated on the curated domain datasets but is used in a different and much larger dataset, i.e. Laion-200M?
    * Will there be OOD shift between the curated domain datasets and Laion-200M?
    * If so, how to make sure the domain classifiers are good?
    * Maybe you can sample a few classified samples in Laion-200M and verify the result by humans.
    * Will adding more domain classifiers and performing an ensemble help?
    * I suggest that the authors include a dedicated subsection in their methodology or results discussing the domain classifier's performance on LAION-200M, including human verification of a sample of classifications and exploration of ensemble methods.

3. Some typos, e.g data points instead of datapoints on lines 76, 82, etc.
4. Overall, I believe this is a technically sound paper with rigorous analysis that supports its main argument.

**Questions:**

I have a few more questions that would love to discuss with the author. However, these questions should not count towards the evaluation of the paper.
1. Do you think the improved generalization comes from merely more data points or a more sophisticated loss or learning paradigm (i.e. contrastive learning and aligning with language)
2. Despite this paper is mainly on CLIP and contrastive vision language model, the main-stream of foundation models are generative models that are instruction-tuned and aligned. Do you think this work and its conclusion will generalized to LLMs? Specifically, one interesting new thing that was not used in CLIP training is the instruction tuning that enables the model to generalize across tasks. In that sense, models will learn to generalize across tasks instead of data points. For example, MLLMs can be trained to generalize to the task of discriminating between any two domains given by the user. Given such ability, will the MLLMs have better OOD generalization?

---

> ### Author Response · Authors · 2024-11-19
>
> Dear Reviewer Gz5f,\
> Thank you for your thoughtful feedback, insightful comments and questions, and for recognizing the strengths of our paper. Below, we address each of your concerns in detail:
>
> **[W1 - Improving the figure]:** Based on your suggestion, we have improved the figure for better readability.
>
> **[W2 - Effectiveness of the Domain Classifier]:** Thank you for raising this important point. The domain classifiers were trained, validated, and tested on randomly sampled subsets of LAION-200M (13K training, 3K validation, and 3K test images), ensuring no distribution shift between their training and evaluation data. While potential generalization error cannot be ruled out, we are confident in the classifier's effectiveness for the following reasons:
>
> 1. **High Precision and Recall:** We prioritized precision by thresholding the classifiers at 98% validation precision on the 3K LAION-200M validation set. The precision and recall metrics on the 3K LAION-200M test set, as well as on 1K samples each from ImageNet and DomainNet test sets, are presented in Tables 1 and 5. These high values demonstrate the classifiers’ reliability. Additionally, accuracy values without thresholding, reported in Tabs. 7 and 8 for LAION-200M and ImageNet/DomainNet sets, further affirm its robustness.
>
> 2. **Visual Verification:** Random samples from the LAION-Natural and LAION-Rendition datasets, derived from deploying the classifiers on LAION-200M, are visualized in Figs. 3, 21, and 22. These samples clearly align with the intended natural and rendition categories, underscoring the classifiers’ efficacy.
>
> 3. **Empirical Validation via Core Results:** Our core results (Sec. 4) show that training models exclusively on natural or rendition subsets leads to strong performance on the corresponding domain but weaker performance on the other. This outcome supports the accuracy of the domain classifiers in retrieving domain-specific samples.
>
> We agree that ensembling different classifiers could further improve performance by leveraging diverse model biases. However, given the already strong performance of our current classifiers, we leave this as an avenue for future work. Finally, in response to your suggestion, we have added a discussion in Sec. A.9 summarizing these points, alongside details in Secs. 3.1, A.1.2, and A.1.5.

---

> > ### Author Response · Authors · 2024-11-19
> >
> > **[Q1 - data distribution vs learning paradigm]:** The improved generalization observed in CLIP primarily stems from the presence of samples from target domains within its training distribution, rather than from the use of a more sophisticated loss function or learning paradigm like contrastive learning or language alignment. Our main findings demonstrate that CLIP's strong performance on rendition domains is largely attributable to domain contamination in the training data. This conclusion aligns with prior work, such as [1], which highlights that CLIP's robustness is predominantly driven by the training distribution itself. Factors like dataset size, language supervision, and contrastive loss have been shown to play minimal roles in the robustness gains. Furthermore, models trained on identical data distributions, irrespective of differences in loss functions (e.g., SimCLR+FT, CLIP, Supervised) or architectures (e.g., backbones and parameter sizes), achieve comparable levels of effective robustness. These findings suggest that training data composition, rather than the specific learning paradigm, is the key driver of generalization performance. We discuss the broader implications of these findings and their validity across dataset sizes in Sec. 6.
> >
> > **[Q2 - Generalization of conclusions to LLMs/MLLMs]:** Thank you for the intriguing questions! If we were to speculate, we believe that for LLMs, domain contamination or the presence of domain-relevant data during training also remains a primary driver of their generalization performance on tasks. Instruction tuning on task-relevant data would likely be essential for generalization to specific tasks. However, identifying and isolating domain- or task-relevant data in the language setting is significantly more challenging, requiring deeper investigation and thoughtful experimental design to enable an interventional analysis like the one presented in our work. Furthermore, prior studies have shown that LLMs often struggle with data that differs slightly from what they likely encountered during training [2, 3]. For MLLMs, we adopt a similarly cautious view: Domain-relevant data appears necessary for generalization. For instance, [4] demonstrates that MLLMs exhibit limited zero-shot generalization to synthetic images, real-world distributional shifts, and specialized datasets, often failing to perform effectively beyond their common training domains. Overall, we believe that generalization challenges akin to those in CLIP, though potentially different in character, will likely persist in LLMs and MLLMs.
> >
> > Thanks again for your questions and feedback. We hope to have addressed all of your concerns.
> >
> > [1] Fang et al, “Data Determines Distributional Robustness in Contrastive Language Image Pre-training (CLIP)” \
> > [2] Mirzadeh et al, GSM-Symbolic: Understanding the Limitations of Mathematical Reasoning in Large Language Models \
> > [3] Dziri et al, Faith and Fate: Limits of Transformers on Compositionality \
> > [4] Zhang et al, On the Out-Of-Distribution Generalization of Multimodal Large Language Models

---

> ### Comment · Reviewer_Gz5f · 2024-11-22
> **Comments from the reviewer**
>
> Dear authors,
>
> Thanks for the comments and rebuttal. I believe all my concerns are addressed. I have been thinking about this topic for a while, which I termed "generalization is probably a hallucination during the era of the foundation model". One interesting point is whether the conclusion would also generalize to LLMs, which most people believe has achieved some sort of generalization.
>
> Best

---

> > ### Author Response · Authors · 2024-12-03
> >
> > Dear Reviewer Gz5f,
> >
> > Thank you for taking the time to review our response and for advocating for our work. We truly appreciate your support. We wholeheartedly agree that extending our conclusions to the LLM space would be fascinating, and we are eager to see and contribute to progress in this area.

---

### Official Review · Reviewer_r683 · 2024-11-04

**Soundness:** 3
**Presentation:** 3
**Contribution:** 3
**Rating:** 6
**Confidence:** 4

**Summary:**

This work focuses on studying the out-of-distribution generalization capabilities of the CLIP model. To do so, authors consider the scope of *rendition* images and propose to use a domain classifier to detect images from this particular domain from the LAION dataset used to train the CLIP model. By training CLIP with this non-contaminated version of the training data, authors showed that the performance of the model on rendition images decayed, indicating that the original model performance on these images could be explained by the presence of such examples in the training set.

**Strengths:**

- S1: This work raises awareness of the importance of curating and performing in-depth analysis of large-scale datasets.

- S2: One of the main contributions of this work is to release two curated partitions of the LAION dataset with non-overlapping domains. This will promote and facilitate future research on single-source domain generalization for foundation models such as CLIP.

**Weaknesses:**

- W1: The motivation for the main question this work aims at answering ("shed light on the limitations of foundation models like CLIP in handling OOD generalization") is not quite clear from the manuscript. In a few words, it seems this work is showing that when CLIP is trained and tested on the same distribution, it performs well, and performance is harmed once a particular domain is removed from the training set and the model is tested on it. What is the exact new insight from this result? Please clarify this in the manuscript.

   - W1.1: In a similar vein, previous work [1] has pointed out that CLIP has limited OOD generalization capabilities and also proposed fine-tuning approaches to mitigate such issues.

- W2: The approach used for data curation relies on collecting human-labeled to train a domain classifier, making it specific to the setting studied in the paper, i.e. rendition vs natural images.

- W3: In Table 2, the authors report the number of natural/ambiguous/rendition images resulting from the filtering with the domain classifier. However, it is possible to see that in most of the cases a considerably high number of images (> 60% in some cases) is deemed as ambiguous. This seems to indicate that either such domain classifier is not a good model to distinguish natural vs rendition images, or that it could be the case that it does not make sense to consider both domains as two distinct sources of data.

- W4: Lack of overall clarity. Some statements such as "We conclusively demonstrate that domain contamination in the training set is what drives CLIP’s robustness", however, if rendition data is present in the training data, then rendition data is no longer OOD with respect to the training distribution.


[1] Shu et al, CLIPood: Generalizing CLIP to Out-of-Distributions, ICML, 2023.

**Questions:**

- Q1: Even after reading the manuscript a few times, the main goal of this work is unclear to me. Previous work on the domain generalization setting has shown that neural networks do struggle to generalize across domains [1], even with powerful regularization and data augmentation approaches to mitigate domain shift. Given that, why exactly would CLIP be expected to generalize well OOD with limited training data? Given that, it seems to me the result/motivation presented in Figure 1-D is exactly what is expected to be given the IID assumption. Can the author kindly elaborate on their motivation?

- Q2: Authors mentioned in line 62 that "Domains, while not always rigorously defined, emerge from collecting data from specific sources
and conditions." However, there is a lot of effort from the machine learning to rigorously define what a domain is. For example, the authors can refer to the work by Ben David et al. [2]

- Q3: Details about the annotation task are missing, please add the following information to the manuscript: how many annotators were considered, inter-rater agreement, how annotators were chosen and compensated, description of the task set up.

- Q4: The authors mentioned in Line 257: "Since human annotators can make mistakes, and LAION-200M’s domain
composition is inherently unknown, we use our domain classifiers to understand it.". However, I am confused by this statement. The domain classifiers the authors refer to were trained on human-labelled examples, which means they can also inherit the mistakes made by annotators, this approach also seems to be potentially affected by human mistakes (as well as the classifier's mistakes).

- **Comment that has not affected my score:** The choice of paper title seems odd to me. The title is quite similar to the reference "In search of lost domain generalization", which can be confusing on its own, but the both papers are also considerably different in the questions asked and methodology, making it even more odd that they have such similar names. Moreover, the title "In search of forgotten domain generalization" does not reflect well the content of this submission as it gives the impression that the contributions are broader than they actually are, since this work only focused on the CLIP model and a specific type of domain shift.


[1] Gulrajani, Ishaan, and David Lopez-Paz. "In search of lost domain generalization." arXiv preprint arXiv:2007.01434 (2020).

[2] Ben-David, Shai, et al. "A theory of learning from different domains." Machine learning 79 (2010): 151-175.

---

> ### Author Response · Authors · 2024-11-18
>
> Dear Reviewer r683,
>
> Thank you for reading our paper and providing feedback. We address your concerns and questions below and aim to clarify our work’s key points:
>
> **[W1 and Q1 - Literature assumes that CLIP generalizes OOD]:** Numerous studies implicitly or explicitly assume that CLIP generalizes OOD, as reflected in the following quotes. We now reflect this sentiment more clearly in the introduction (Lines 80–88).
> - “zero-shot transfer is more an evaluation of CLIP’s robustness to distribution shift and domain generalization” [1]
> - “several studies suggested that some Vision-Language Models (VLMs) such as the CLIPs, exhibit OoD generalization” [2]
> - “VLP (Vision-Language Pretraining) models not only show good performance in-distribution (ID) but also generalize to out-of-distribution (OOD) data” [3]
> - “models such as CLIP, ALIGN, and BASIC have recently demonstrated unprecedented robustness on a variety of natural distribution shifts” [4]
> - “These models (CLIP, GLIP, etc.) have demonstrated enormous potential in, a wide range of downstream applications… especially for the out-of-distribution (OOD) samples” [5]
> - and, finally, the paper you cited notes that “CLIP achieves OOD generalization in a task-agnostic way” [6].
>
> **[W1 and Q1 - Clarification of core insight]:** As noted above, many works attribute CLIP’s high performance on OOD benchmarks to inherent robustness or emergent properties of large training datasets. However, our findings challenge this interpretation. While, as you point out, under the IID assumption, models are expected to perform poorly when an entire domain is excluded from training, this principle has often been overlooked when evaluating foundation models like CLIP. Our key contribution is to demonstrate conclusively that CLIP’s strong performance on these benchmarks stems from domain contamination in its training set, not intrinsic robustness or OOD generalization (see Sec. 3.3 and Sec. 4). To reiterate, when models trained on uncurated, web-scale datasets excel on fixed OOD benchmarks like ImageNet-Sketch, it’s often assumed they can generalize to arbitrary shifts [1–6]. However, if the training and test data are not truly distinct, these evaluations reflect in-domain performance rather than genuine OOD robustness. In Sec. 4, we demonstrate this flaw by ensuring the test data is genuinely OOD—removing similar/in-domain samples from the training set—and observing that CLIP fails to maintain its performance. This highlights that CLIP’s OOD performance on renditions critically depends on training data containing rendition images. Our work thus underscores a significant evaluation flaw, serving as a cautionary tale about the need for rigor in OOD evaluations and tempering expectations regarding CLIP’s true generalization capabilities. To better emphasize this core insight, we have revised the main text (Lines 80–88) and included a more detailed discussion (Sec. 6 first paragraph).
>
> **[W1.1 - Regarding CLIPood]:** Our work explicitly examines the ‘zero-shot generalization’ capabilities of CLIP models. In this context, even CLIPood [6], observes that *“vision-language pre-trained models demonstrate impressive zero-shot learning performance and outperform models trained from only labeled images, which reveals a promising approach toward OOD generalization”* and that *“CLIP achieves OOD generalization in a task-agnostic way.”* [6] focuses on the degradation of this perceived generalization on certain tasks when adapting to another specific set of tasks, distinct from the problem we investigate in this work.
>
> **[W2 - Reliance on human-labeled data]:** To our knowledge, there is no automated or principled method to reliably differentiate between natural and rendition domains. Moreover, as highlighted in Sec. 3.3, Tab. 9, and Appx. A.1.6, even widely-used single-domain evaluation datasets (e.g., DomainNet, ImageNet-Sketch) are contaminated with samples from other domains, making them unsuitable as training datasets for the domain classifier. Additionally, since LAION lacks domain labels and required domain classification for our study, we had to manually label data from the dataset to ensure the domain classifier’s accuracy and generalizability. That said, if clearly demarcated datasets—human-labeled or otherwise—exist for other domains, our methodology can be seamlessly applied to those settings, and we expect our conclusions to also hold there.

---

> ### Author Response · Authors · 2024-11-18
>
> **[W3 - Imperfect domain classifier]:** We acknowledge the high percentage of images classified as ambiguous, caused by two design choices: Our goal was to isolate images that are *strictly* natural or *strictly* renditions. We, therefore, define as ambiguous any image containing stylized and natural elements, such as photos of tattoos, or objects with stylized logos (see Appx. A.1.1). Additionally, we prioritized high precision by deploying our classifier at 98% validation precision, which inevitably led to more false negatives labeled as ambiguous. Our conservative strategy ensures that the images identified as natural or renditions are highly reliable. Overall, the large number of samples our classifier can annotate for each domain and the existence of distinct datasets, such as ImageNet-Val and ImageNet-Sketch, support the validity of treating natural and rendition images as distinct data sources. Nevertheless, we agree that further exploring the characteristics and impact of borderline or ambiguous examples could provide valuable insights into generalization. We leave this exercise for future work.
>
> **[W4 - Lack of clarity]:**  We completely agree with your observation *“if rendition data is present in the training data, then rendition data is no longer OOD with respect to the training distribution.”* This is precisely the core point of our work. As referenced in [1–6], many works implicitly or explicitly assume that CLIP demonstrates OOD generalization. However, our findings challenge this assumption, showing that CLIP primarily achieves in-domain generalization due to domain contamination in its training data. To clarify this in the text, we have revised ambiguous statements like *“domain contamination drives CLIP’s robustness,”* to *“CLIP’s apparent OOD robustness on standard OOD benchmarks like ImageNet-Sketch is often an artifact of overlapping domain data, rather than genuine OOD generalization.”* This adjustment will ensure the language better conveys our central message.
>
> **[Q2 - Defining domain differences]:** Thank you for highlighting this excellent reference. We intended to emphasize that measuring domain overlap can be challenging in practice, especially on web-scale data. We will cite the work by Ben-David et al. and revise the language to: "Domains, while often challenging to quantify in practice [7], emerge from collecting data from specific sources and conditions."
>
> **[Q3 - Annotation details]:** Thank you for your suggestion. The details of the labeling procedure, including the annotators and quality assurance measures, are provided in Sec. 3.1, Appendix Sec. A.1.1, and A.1.4. In response to your recommendation, we have added further specifics. We appreciate your feedback in helping us improve the clarity and completeness of this section.
>
> **[Q4 - Bias due to labeling]:** Thank you for highlighting this important nuance. You are correct that biases in human labeling can propagate to the classifier and potentially influence the results. However, by deploying our classifiers at high precision, we minimize domain contamination in the generated datasets, ensuring that our subsequent conclusions remain robust and valid. We have revised the sentence to: “Since human annotation for several million samples is expensive and prone to error, we rely on high-precision domain classifiers to filter the datasets.” This updated phrasing more accurately reflects our approach and its limitations. We have also added a paragraph regarding this to our extended discussion section (Sec. A.9).
>
> **[Q5: On the title]:** Despite being a longstanding open problem, ever since the advent of web-scale datasets, we believe that the community at large has forgotten to channel efforts into principally studying domain generalization, as was done in the ImageNet/-R/-Sketch era. We, therefore, went with this title but are open to following the reviewers’ consensus and changing it.
>
> We hope our detailed responses have thoroughly addressed your questions and clarified our contributions. We kindly request you to reevaluate our work in light of this clarification and the feedback from other reviewers.
>
> [1] Radford et al, “Learning Transferable Visual Models From Natural Language Supervision”
>
> [2] Abbasi et al, “Deciphering the Role of Representation Disentanglement: Investigating Compositional Generalization in CLIP Models”
>
> [3] Nguyen et al, “SAFT: Towards Out-of-Distribution Generalization in Fine-Tuning”
>
> [4] Fang et al, “Data Determines Distributional Robustness in Contrastive Language Image Pre-training (CLIP)”
>
> [5] Li et al, “Distilling Large Vision-Language Model with Out-of-Distribution Generalizability”
>
> [6] Shu et al, “CLIPood: Generalizing CLIP to Out-of-Distributions”
>
> [7] Ben-David et al, “A theory of learning from different domains”

---

> > ### Comment · Reviewer_r683 · 2024-11-25
> > **Updating my score**
> >
> > Dear authors,
> >
> > I appreciate your thorough rebuttal and updated my score from 3 to 6 to reflect my new evaluation of your work.
> >
> > The clarifications regarding the main question investigated in this work were very helpful and I found that the revised introduction captured the new points brought by the authors in the rebuttal.
> > Even though most of my concerns were addressed by the rebuttal, W2 still remains. The approach is still tailored to the case where renditions vs natural images are being investigated, and the conclusions are specific to the CLIP model.
> >
> > Regarding the title (which hasn't influenced my score), I believe it should reflect the fact that this work is focused on the CLIP model and investigating a particular kind of domain shift. The current title gives the impression that the scope is much broader.

---

> > > ### Author Response · Authors · 2024-11-26
> > >
> > > Dear Reviewer r683,
> > >
> > > Thank you for your valuable feedback and adjusting your score. We acknowledge that our experiments are limited to the natural/rendition domains and CLIP. We discuss each of these limitations in detail in Sec. 6, which we have also updated in the course of this rebuttal to better reflect why our results will nonetheless be relevant for other models and domains.
> > >
> > > In a nutshell, prior work has demonstrated conclusively that data is the driving factor of a perceived domain generalization capability in large models across loss functions and architectures. Some of these methods are also trained on LAION, but generally no public large training set accounts for domain contamination to commonly used test sets. Consequently, we see no reason to believe that other methods will generalize better.
> > >
> > > Collecting large-scale test sets and training these models from scratch is costly, and we believe it is the responsibility of authors who propose new, purportedly more robust methods to evaluate them properly. We release our datasets for the natural/rendition domain to facilitate this and publish our methodology to enable the curation of similar test sets for other domains. We hope our demonstration using CLIP can serve as a wake-up call to the community and drive future research in this area.

---

### Official Review · Reviewer_HGzr · 2024-11-05

**Soundness:** 4
**Presentation:** 4
**Contribution:** 4
**Rating:** 8
**Confidence:** 4

**Summary:**

This paper investigates if the recent purported gains in generalization reported for Large scale vision language models come from true generalization beyond the data distribution, or from data contamination. To this end, they present a methodology for constructing disjoint subsets of the LAION dataset, which serve as OOD counterparts to ImageNet and DomainNet. This ensures no data contamination, and allows investigation of true OOD generalization capabilities of these models. Their main finding is like CLIP like models to not generalize well when such disjoint sets are constructed carefully. This suggests that while these models appear to generalize, they are merely performing well because they have already seen data close to the test set.

**Strengths:**

Here’s a refined take on the strengths for your review:

1. **Extremely important problem**: This paper addresses a critical issue in machine learning—distinguishing true OOD generalization from data contamination—by revisiting fundamental challenges of OOD generalization in the context of large-scale vision-language models. Their work highlights the limitations of current evaluation practices, underscoring the importance of ensuring genuine robustness and generalization capabilities.

2. **Impactful finding**: The finding that models like CLIP may not truly generalize when faced with rigorously OOD data is significant, as it challenges current assumptions about the robustness of these models. This insight has substantial implications for future work, suggesting that existing performance metrics may overstate the true generalization abilities of these models.

3. **Useful datasets**: By creating the LAION-Natural and LAION-Rendition datasets—both strictly OOD to ImageNet and DomainNet—this work offers a valuable resource for researchers focused on OOD robustness. These carefully constructed subsets provide a way to assess and enhance model generalization on web-scale datasets without contamination.

4. **Useful methodology**: The methodology for creating disjoint, OOD subsets of LAION is a wel established approach in the field, and a solid blueprint for other research seeking to evaluate OOD performance rigorously.

5. **Results/Figures/Tables are very high quality**: The results, figures, and tables presented in the paper are meticulously crafted, enhancing the clarity and interpretability of the findings. They effectively illustrate the study's key points and provide a solid foundation for future discussions on OOD generalization and robustness evaluation.

**Weaknesses:**

1. **Missing Literature:** The field of OOD (out of distribution) generalization is very rich, and there are several related papers that are not cited. It would be nice to have this work related to existing works in the field (See below)

### References

a. Liu, J., Shen, Z., He, Y., Zhang, X., Xu, R., Yu, H. and Cui, P., 2021. *Towards out-of-distribution generalization: A survey*. arXiv preprint arXiv:2108.13624.

b. Koh, P.W., Sagawa, S., Marklund, H., Xie, S.M., Zhang, M., Balsubramani, A., Hu, W., Yasunaga, M., Phillips, R.L., Gao, I. and Lee, T., 2021, July. *Wilds: A benchmark of in-the-wild distribution shifts*. In International conference on machine learning (pp. 5637-5664). PMLR.

c. Madan, S., Henry, T., Dozier, J., Ho, H., Bhandari, N., Sasaki, T., Durand, F., Pfister, H. and Boix, X., 2022. *When and how convolutional neural networks generalize to out-of-distribution category–viewpoint combinations*. Nature Machine Intelligence, 4(2), pp.146-153.

d. Gulrajani, I. and Lopez-Paz, D., 2020. *In search of lost domain generalization*. arXiv preprint arXiv:2007.01434.

e. Madan, S., You, L., Zhang, M., Pfister, H. and Kreiman, G., 2022. *What makes domain generalization hard?* arXiv preprint arXiv:2206.07802.

f. Arjovsky, M., Bottou, L., Gulrajani, I. and Lopez-Paz, D., 2019. *Invariant risk minimization*. arXiv preprint arXiv:1907.02893.

g. Arjovsky, M., 2020. *Out of distribution generalization in machine learning* (Doctoral dissertation, New York University).

2. **Smaller training dataset size:** One of the key claims in papers presenting vision language models is the idea of Scaling Laws---behavior emerges from large scale datasets which cannot be seen in smaller models. Here, CLIP was trained on 19,000 images which is too small and thus it remains unclear if such generalization errors will continue in extreme dataset sizes. If nothing else, it is worth discussing this point in the paper.

**Questions:**

1. Is there a way of automating the dataset slip so as to train the model on a large scale dataset?

2. Would it make sense to generate "un-natural" images i.e. data that does not relate to real world categories to explore this further. For instance, maybe texture images generated from explicitly disjoint data distributions?

---

> ### Author Response · Authors · 2024-11-18
>
> Dear Reviewer HGzr,\
> Thank you for your insightful feedback and for acknowledging the strengths of our work. We address your comments in detail below:
>
> **[W1 - Missing literature]:** We appreciate you bringing these relevant works to our attention. We have incorporated them in the first paragraph of the Related Work section (Sec. 2) on Lines 132 and 133.
>
> **[W1 - Smaller training data size]:** To clarify, the training dataset size of 19,000 refers to our domain classifier (Line 205 in Sec. 3.1). The CLIP models used in our primary experiments (Sec. 4) were trained on datasets ranging from 4M to 57M in size (see Sec. 3.4, Tab. 12, x-axis of Fig. 4). We now also include this information in Fig. 3 to make this point clearer. Additionally, we direct your attention to Lines 500–507 in Section 6, where we discuss the validity of our conclusions for larger dataset sizes. Based on your suggestion, we have further elaborated on our reasoning for the scalability of our conclusions in the second paragraph of Sec. 6:
> > The disparity in `relative corrected accuracy' shown in Fig. 4 remains stable across dataset sizes from 4M to 57M. Similarly, effective robustness illustrated in Fig. 5 is influenced by the training distribution rather than the dataset size, which is also supported by findings in previous works (Miller, 2021, Fang, 2022, Mayilvahanan, 2023). Lastly, CLIP’s performance scales predictably across domain mixtures as shown in Sec. 5. Overall, we see no indication that our results should not transfer to larger scales.
>
> **[Q1 - Is there a way to automatically curate the data splits?]:** Yes! This is already part of our approach: We labeled 19,000 datapoints to train the domain classifier as described above, and used this classifier to identify 57M/16M natural/rendition images in LAION-200M. The same classifier could be used to label the domain of images from a larger pretraining distribution, if available.
>
> **[Q2 - Generate “Unnatural” Images]:** This is an excellent idea, and was the motivation behind the ImageNet-C dataset. The specific corruptions employed in ImageNet-C are not OOD with respect to LAION, but other novel corruptions might be possible. We leave this analysis for future work.
>
> Thanks again for your questions and feedback. We hope to have addressed all of your concerns.

---

### Author Response · Authors · 2024-12-03

Dear Reviewers,

We sincerely thank you for your thoughtful feedback and are delighted by your recommendations for acceptance. Your comments and praise for our work as addressing an **“extremely important/valuable problem”** (Rev. HGzr, Gz5f), conducting **“in-depth analysis of large-scale datasets”** (Rev. r683), and delivering **“impactful/credible findings”** (Rev. HGzr, 8EXX) are deeply appreciated. Additionally, we are grateful for your recognition of our **“useful datasets/useful methodology”** (Rev. HGzr, r683) and the **“high-quality results/figures/tables”** (Rev. HGzr) achieved through **“rigorous analysis/extensive experimentation”** (Rev. HGzr, Gz5f, 8EXX).

In response to your valuable suggestions and questions, we have implemented the following improvements in our paper:
- **Section 1:** Enhanced to clarify that the literature implicitly or explicitly assumes CLIP generalizes OOD, and contextualized our key findings in this context (Rev. r683, 8EXX).
- **Literature Review:** Added several relevant citations to strengthen and position our work within the existing body of research (Rev. HGzr, r683).
- **Figures 3 and 6:** Refined for better clarity, with improved captions for ease of understanding (Rev. HGzr, Gz5f).
- **Section 3.1 & Appendices A.1.1 and A.1.4:** Expanded to provide greater detail on our domain classifiers, labeling pipeline, and procedures (Rev. r683).
- **Section 6 and New Section A. 9:** Improved the discussion section to address points raised by reviewers. Added an extended discussion section to elaborate on the contextualization of our results, validity across other architectures and losses, potential labeling bias, and object class distribution considerations (Rev. HGzr, r683, Gz5f, 8EXX).

We once again thank all reviewers for engaging deeply with our work, recognizing its contributions, and advocating for its acceptance. Your insights have been instrumental in further strengthening our paper.

---

### Meta-Review · Area_Chair_wfMu · 2024-12-15

**Metareview:**

The paper addresses a critical and timely issue of out-of-distribution (OOD) generalization in the era of foundation models.
It challenges the common assumption that models like CLIP inherently generalize OOD, providing evidence that their strong performance often stems from domain contamination in training datasets.

The paper introduces two rigorously curated datasets, LAION-Natural and LAION-Rendition, which enable meaningful evaluation of OOD generalization. These datasets will have substantial utility for future research in improving model robustness and generalization.

Overall, this paper is well-written, addresses a significant problem, provides high-quality datasets and experiments, and has received strong support from reviewers. Given its contributions to OOD evaluation and potential to inspire further research in robust model development, it is suggested that this paper be accepted.

**Additional Comments On Reviewer Discussion:**

- R HGzr was concerned about the training dataset size, which the authors clarified in their rebuttal.
- R r683 was not clear with the goal/insight of this paper. The authors did a good job of clarifying them in the rebuttal and achieved a score raise (from 3 to 6) from r683.

---

### Decision · Program_Chairs · 2025-01-22

Accept (Spotlight)